# A flexible generative algorithm for growing in silico placentas

**Diana C. de Oliveira**[1]*, **Hani Cheikh Sleiman**[1], **Kelly Payette**[2,3], **Jana Hutter**[2,3,4], **Lisa Story**[5], **Joseph V. Hajnal**[2,3], **Daniel C. Alexander**[6], **Rebecca J. Shipley**[1], **Paddy J. Slator**[7,8,6]

**1** Department of Mechanical Engineering, University College London, London, United Kingdom, **2** Centre for the Developing Brain, School of Biomedical Engineering and Imaging Sciences, King's College London, London, United Kingdom, **3** Biomedical Engineering Department, School of Biomedical Engineering and Imaging Sciences, King's College London, London, United Kingdom, **4** Smart Imaging Lab, Radiological Institute, University Hospital Erlangen, Erlangen, Germany, **5** Department of Women and Children's Health, School of Life Course Sciences, King's College London, London, United Kingdom, **6** Centre for Medical Image Computing and Department of Computer Science, University College London, London, United Kingdom, **7** Cardiff University Brain Research Imaging Centre, School of Psychology, Cardiff, United Kingdom, **8** School of Computer Science and Informatics, Cardiff University, Cardiff, United Kingdom

\* d.oliveira@ucl.ac.uk

**Data Availability Statement:** The full source code of the toolbox and examples of feto-placental vascular structures are available from Zenodo (https://doi.org/10.5281/zenodo.10557280).

## Abstract

The placenta is crucial for a successful pregnancy, facilitating oxygen exchange and nutrient transport between mother and fetus. Complications like fetal growth restriction and pre-eclampsia are linked to placental vascular structure abnormalities, highlighting the need for early detection of placental health issues. Computational modelling offers insights into how vascular architecture correlates with flow and oxygenation in both healthy and dysfunctional placentas. These models use synthetic networks to represent the multiscale feto-placental vasculature, but current methods lack direct control over key morphological parameters like branching angles, essential for predicting placental dysfunction.

We introduce a novel generative algorithm for creating *in silico* placentas, allowing user-controlled customisation of feto-placental vasculatures, both as individual components (placental shape, chorionic vessels, placentone) and as a complete structure. The algorithm is physiologically underpinned, following branching laws (i.e. Murray's Law), and is defined by four key morphometric statistics: vessel diameter, vessel length, branching angle and asymmetry. Our algorithm produces structures consistent with *in vivo* measurements and *ex vivo* observations. Our sensitivity analysis highlights how vessel length variations and branching angles play a pivotal role in defining the architecture of the placental vascular network. Moreover, our approach is stochastic in nature, yielding vascular structures with different topological metrics when imposing the same input settings. Unlike previous volume-filling algorithms, our approach allows direct control over key morphological parameters, generating vascular structures that closely resemble real vascular densities and allowing for the investigation of the impact of morphological parameters on placental function in upcoming studies.

**Funding:** The research leading to these results has received funding from the EPSRC (grant no. EP/V034537/1) to DA, which partly funds DO and fully funded PS and KP, and from the Wellcome Leap In Utero award to DA, which partly funds DO and PS. HCS was partly funded by the CRUK (C44767/A29458) and OncoEng project (EP/W007096/1), both awarded to RJS. We also acknowledge funding from the UKRI FLF (MR/T018119/1) and DFG Heisenberg (502024488) to JH, as well as core funding from the NIHR Biomedical Research Centre based at Guy's and St Thomas' NHS Foundation Trust and KCL and at UCLH. We also acknowledge core funding from the Wellcome/EPSRC Centre for Medical Engineering (WT203148/Z/16/Z). The funders had no role in study design, data collection and analysis, decision to publish, or preparation of the manuscript.

**Competing interests:** The authors have declared that no competing interests exist.

## Author summary

The placenta is important in ensuring a healthy pregnancy by facilitating the exchange of oxygen and nutrients between the mother and the fetus. Disturbances of placental function are often associated with abnormalities in the placental vascular structure, and detecting these issues early on is crucial. To understand the connection between placental vascular architecture, blood flow, and oxygenation, computational models have been used. These use synthetic networks which lack precise control over crucial morphological parameters, such as branching angles, essential for predicting placental dysfunction. Our contribution is a new approach that allows for the creation of virtual placentas that closely resemble real vascular characteristics. It enables users to customize the feto-placental vascular architecture at various levels, including individual components like placental shape, chorionic vessels, and placentone, as well as the complete structure. The flexibility of this pipeline opens the door for investigating the direct impact of morphological parameters on placental function.

## Introduction

The placenta is vitally important for a successful pregnancy and influences the lifelong health of both the child and the mother [1, 2]. Fetal and maternal blood flow separately through a complex branching system that maximises oxygen exchange between mother and fetus to enable appropriate fetal growth (Fig 1). At a macro scale, the feto-placental vasculature encompasses the fetal umbilical vessels, connecting the placenta to the fetus and carrying either deoxygenated fetal blood (via the umbilical arteries) or oxygenated fetal blood (via the umbilical vein). These umbilical vessels give rise to chorionic vessels, which extend through the

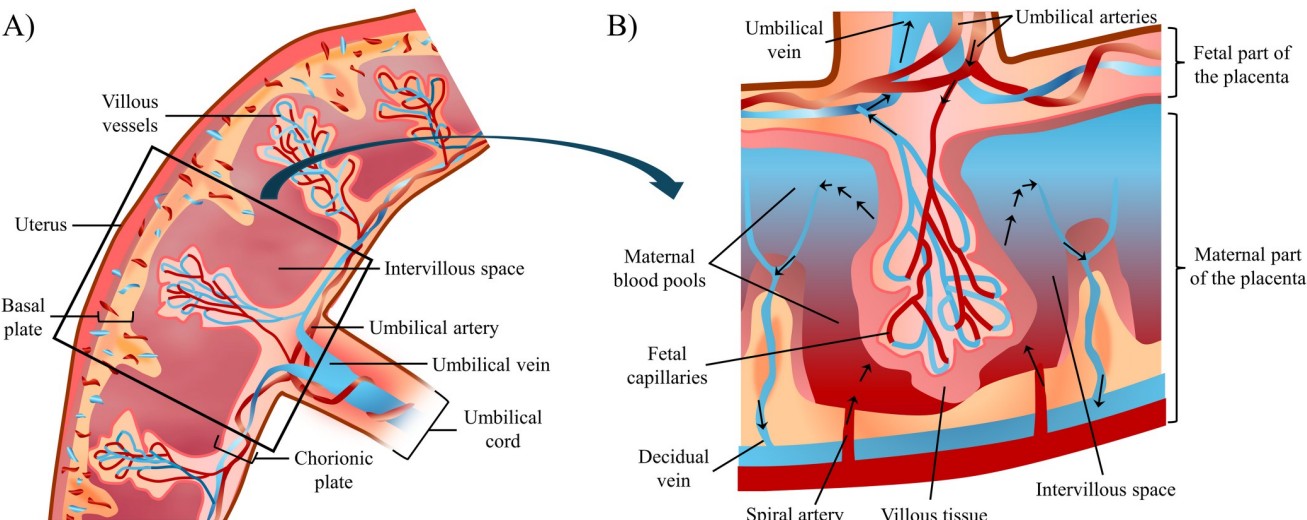

**Fig 1. Schematic representation of the placenta and feto-placental vascular structures.** (A) Placental macroscopic components and associated structures, including the chorionic and basal plates and the umbilical cord. (B) A functional lobule of the feto-placental circulation is shown in detail: exchange occurs in the IVS, most specifically in the maternal blood pools arising from the spiral arteries. Oxygenated blood enters the IVS through the spiral arteries. Deoxygenated blood from the fetal side enters the placental parenchyma through the umbilical arteries, via the villous tree up to the capillaries, where it is oxygenated before returning to the fetus in the umbilical vein. Oxygen-depleted blood then leaves the IVS through the decidual veins located in the septa between the IVS.

chorionic plate and supply smaller-scale villous trees. At a microscale, the villi further branch into intricate capillary pathways that facilitate the aforementioned gas exchange. The villous trees are considered functional units of the placenta, and define the intervillous space (IVS), where maternal blood enters to provide nutrients and gas exchange. The spiral arteries arising from the uterus yield low-velocity, low-pressure blood flow in the IVS, optimising villous tree perfusion [3–5]. This oxygenated blood then becomes deoxygenated as it passes through the villous trees, exiting the IVS through decidual veins [5–7]. The spatial arrangement of the feto-placental vasculature is influenced by various structural and functional factors, including placental size and shape, or blood flow velocity within the IVS.

Many pregnancy complications, including fetal growth restriction (FGR) and pre-eclampsia, are associated with placental dysfunction, including abnormalities of placental vascular structures and function [8–10]. Quantitative measurements of placental health during pregnancy can improve detection and monitoring of pregnancy complications. However, despite the emergence of promising techniques such as placental magnetic resonance imaging (MRI) [4, 11–14], non-invasive observation of the placenta during pregnancy remains highly challenging. There is hence poor understanding of the links between pregnancy complications and underlying placental structure and function [15–17]. While animal models allow insights into specific pathophysiological mechanisms, they do not allow us to study the unique aspects of the development, adaption, structure and function of the human placenta [18]. On the other hand, biophysical models can provide insights into placental structure and function as they employ vascular structures as inputs to predict functional parameters relevant to the performance of the human placenta [19]. Such models can simulate blood flow patterns and solute transport within the placental vasculature, helping to identify regions with optimal or compromised perfusion [20, 21], and underlying the transport of nutrients and oxygen across the placental barrier [22, 23]. Therefore, they allow testing of different structural-functional hypotheses in an *in silico* setting to better understand human pregnancy [24–26].

Realistic *in silico* placental geometries required for biophysical modelling have been previously acquired via detailed imaging of post-delivery placentas using techniques such as micro-computed tomography [27], photoacoustic imaging [28], and MRI [29]. However, such techniques are costly and time consuming and require a trade-off between spatial coverage and spatial resolution [30]. Moreover, while the placenta has a round, curved shape determined by the curvature of the uterine wall, post-delivery it appears flatter and more irregular, and may also show signs of physical damage or detachment from the uterine wall [31] caused by the birthing process. This means that *in vivo* and *ex vivo* feto-placental vasculature differs and geometrical details captured post-delivery may not be entirely representative of the *in vivo* placenta.

Several modelling approaches have therefore utilised growing algorithms to generate plausible *in silico* feto-placental vascular structures, including the definition of an outer placental surface, the growth of chorionic vessels and the generation of villous trees. This has typically been achieved through the use of volume filling algorithms, where segments grow towards user-defined seed points based on heuristic constraints, such as maximum branching angles and fractional distances towards seed point centres of mass [20, 21, 32].

A major limitation of volume-filling algorithms is that multiple vital morphological parameters, such as the actual branching angle, number of branching generations and length-to-diameter ratios are not directly controllable, but rather are emergent properties of the algorithm [33]. It is therefore difficult, and potentially impossible, to directly control a single morphological parameter, and study its effect on the flow and transport properties of the placenta. To test hypotheses about the influence of structural differences in dysfunction, we need to control these parameter settings while generating *in silico* models. Given the uncertainty associated with morphological parameters arising from different acquisition techniques (*in vivo* and

*ex vivo*), sensitivity analyses of the effect of key parameters on topological metrics becomes paramount to quantify the impact of uncertainty in input parameters on vascular structures, providing informed insights into the relationships between inputs, structure and function.

In this paper we address these limitations with a flexible generative algorithm for growing *in silico* placentas where all the salient morphological parameters are user controlled. We derive sensible default values for these parameters from the literature and assess their influence on vasculature topology via a sensitivity analysis. This is crucial, given the uncertainty in placental structural parameters arising from different assessment techniques. The algorithm and exemplar *in silico* placental structures are made freely available at [34]. These can underpin subsequent placenta modelling including, flow, oxygen and nutrient transport, and simulations of MRI signals.

## Background: Morphology of the placenta and feto-placental vasculature

### Key features and association with pathology

Several morphological features of the placenta and its vascular structure determine placental efficiency and the appropriate growth of the fetus. These features need to be accounted for in a generative algorithm for the feto-placental vasculature at a multi-scale level (Fig 1).

At a macroscopic scale, the placenta is usually described as round or oval, with high variability in size and shape (with e.g. average thickness of placental disk varying from 0.951 to 3.095 cm and the placental surface diameter from 14.933 to 33.499 cm within one clinical dataset [35]) [36, 37]. From a structural point of view, placental shape and size are usually viewed as markers of placental health and efficiency, as well as perinatal outcome [35]. For example, alterations to placental shape observed post-delivery [6, 9, 38] and with *in vivo* MRI [39], such as smaller total placental volumes and smaller IVS volume, have been associated with pathologies such as FGR and pre-eclampsia, which impact appropriate fetal growth [6, 9, 38, 39].

The chorionic and basal plates form the fetal and maternal sides of the placenta, respectively, with the umbilical vessels arising from the umbilical cord forming the chorionic plate. Variability in vasculature specifics (e.g. spread of chorionic vessels) influence the shape of the chorionic plate and the overall placental shape [37, 40]. Indeed, Salafia *et al.* found a connection between abnormal chorionic plate shapes and altered vascular structures, in turn associated with reduced placental efficiency [37]. The position of the placental umbilical cord insertion is also considered a marker of perinatal outcome. For example, eccentric cords, which yield largely asymmetric chorionic vasculatures, have been associated with placental insufficiency [36, 41]. This means that both the position of the umbilical cord insertion and the structure and dispersion of the chorionic vessels over the chorionic plate affect blood flow supply to and from the smaller-scale villous trees and ultimately the capillary vessels, effectively dictating placental efficiency [36, 41].

The placental basal surface consists of functional lobules separated by grooves, representing placental septa. Each functional lobule usually aligns with fetal villous trees and corresponding decidual vessels. During early pregnancy, maternal spiral arteries undergo remodelling caused by placental cells (trophoblasts), transforming into wide, funnel-like structures [3, 4, 6]. This process facilitates low-velocity, low-pressure blood flow in the IVS [3–5]. Disruptions in spiral artery (SA) remodelling are linked to complications such as placental insufficiency and pre-eclampsia [4, 42]. For instance, incomplete remodelling can lead to smaller SA vascular lumens, elevated, jet-like blood flow rates into the IVS and impacting villous tree maturation over gestation [3]. SA mega-jets, linked to higher flow rates, have been associated with sparser

villous trees and lower vascular density in the placenta [3], though their impact on placental pathology remains unclear.

The fetal villous trees are asymmetric branching structures composed of numerous villi, extending from the chorionic plate into the IVS. The distal ends of these trees are composed by capillary pathways [20, 43]. Ongoing vessel maturation throughout gestation yields denser vascular trees, facilitating the exchange of nutrients, oxygen, and waste products between maternal and fetal circulations [4]. In pathological conditions such as FGR or pre-eclampsia, there are notable alterations in the structure of villous trees, including abnormalities in the size, shape, and density. For example, sparse and elongated vascular networks with decreased villous volumes and surface areas are usually observed in FGR [4, 38, 44]. The spatial distribution and structural features of villous trees are, therefore, key to placental efficiency.

## Morphological parameters

A range of morphological parameters describes the placental surface and the vascular network of blood vessels representing the feto-placental vasculature. Here we focus on structural metrics that will be directly used to create *in silico* placentas and feto-placental vasculatures with our generative algorithm (see Fig 2). For a certain vessel $B$, defined by start and end nodes $i$ and $j$, we can also define the length-to-diameter ratio (*ltd*) as

$$ltd_B = \frac{l_B}{d_B}, \tag{1}$$

where $d_B$ and $l_B$ are the vessel diameter and length, respectively [30]. Using the same notation as in Fig 2B, the 3-dimensional (3D) branching angle between vessel segments $ji$ and $ig$ is given by

$$\theta_{jig} = \arccos \frac{(C_j - C_i) \cdot (C_g - C_i)}{||C_j - C_i|| \cdot ||C_g - C_i||}, \tag{2}$$

where $C_i$, $C_j$, $C_g$ are the spatial positions of vascular nodes $i, j, g$ in a universal coordinate system, respectively [30]. Planar branching angles, also represented in Fig 2B, have been previously used to characterise fetal villi [10]. Planar branching angle values can then be used to approximate 3D branching angles when generating the feto-placental vasculature.

## Model

In this section, we derive our generative algorithm whilst explaining the underpinning rationale and definitions (Algorithm overview and general notation-Capillary representation and venous system). We also describe how key metrics are computed (Morphological metrics computed), our sensitivity analysis on the influence of algorithm input parameters on key topological metrics (Topological assessment and sensitivity analysis), stochastic effects (Stochastic effects) and generated feto-placental vasculatures (Generating healthy and dysfunctional feto-placental vasculatures). A MATLAB implementation of our algorithm, including examples of synthetic placental structures generated, is available at [34].

We introduce our algorithm in stages. We focus on the generation of arterial vessels while still accounting for empty space to be occupied by venous vessels. For each stage, we state the generative procedure and the required user-defined parameters, and the motivation underlying our choices. We specify fixed parameters, but reserve the details of user-defined parameters and literature-based estimates until the Methods section (Tunable parameter definitions and suggested ranges). In-depth algorithm implementation details such as fine-tuning of vessel spatial distribution (Computational implementation: Fine-tuning spatial distribution of feto-

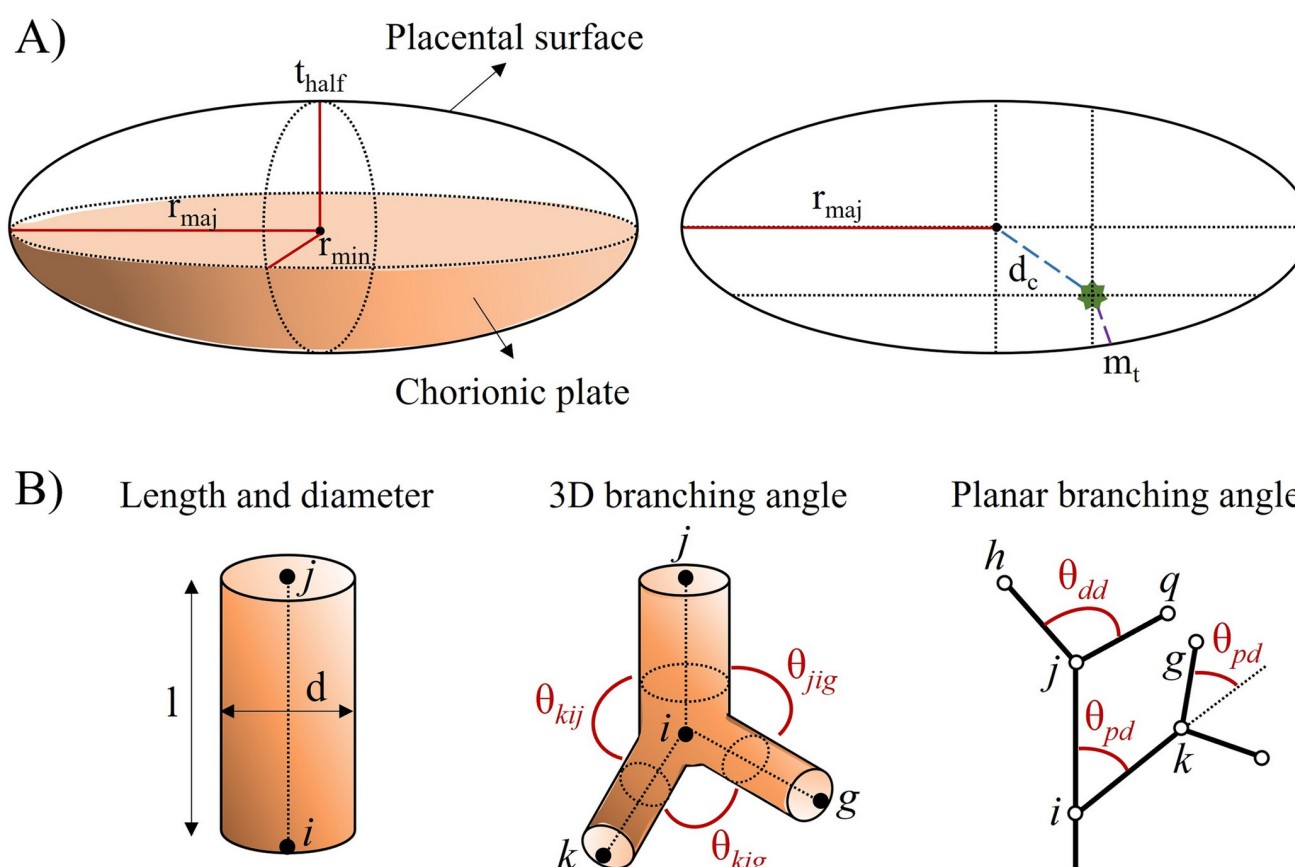

**Fig 2. Schematic representation of important placental morphological parameters.** (A) Typical descriptions of chorionic surface shape and size rely on ellipsoid representations characterised by major ($r_{maj}$) and minor ($r_{min}$) radii [36, 37] and half the placenta thickness ($t_{half}$), as represented in a side view of the placental surface (left side) [20]. Other relevant parameters represented in an axial view of the chorionic plate (right side) include the position of the umbilical cord insertion in the chorionic plate (green star), determined by the distance of the insertion from the ellipsoid centre ($d_c$) and the minimum distance between the insertion and the periphery of the chorionic plate ($m_t$) [36]. (B) Blood vessels are typically characterised by length ($l$) and diameter ($d$) (left side) and 3D branching angles ($\theta$) (middle) [20, 21]. Planar branching angles are defined upon vessel branching properties (right side): The parent-daughter branching angle of a certain segment is defined as the angle of a daughter segment from its parent's axis (e.g. $\theta_{pd}$), while the daughter-daughter branching angle is represented as the angle between two daughter segments (e.g. $\theta_{dd}$).

placental vasculature) and vessel-to-vessel intersection checks (Computational implementation: Branch intersection checks) are provided in the Methods section. For those solely interested in grasping the algorithm's overarching concepts, we recommend consulting section Algorithm overview and general notation, Fig 3.

## Algorithm overview and general notation

Our generative algorithm is implemented in MATLAB (MATLAB, R2021b, 9.11, The Math-Works Inc., Natick, MA, USA). In the following sections, we describe:

- The notation used to define the feto-placental vasculature as trees (Tree notation) and branching rules imposed (Branching rules)

- The definition of placental shape and placental surface mesh, as well as cord insertion (Step-by-step description of placental size and shape-Step-by-step description of placental mesh and cord insertion)

## Type of geometry

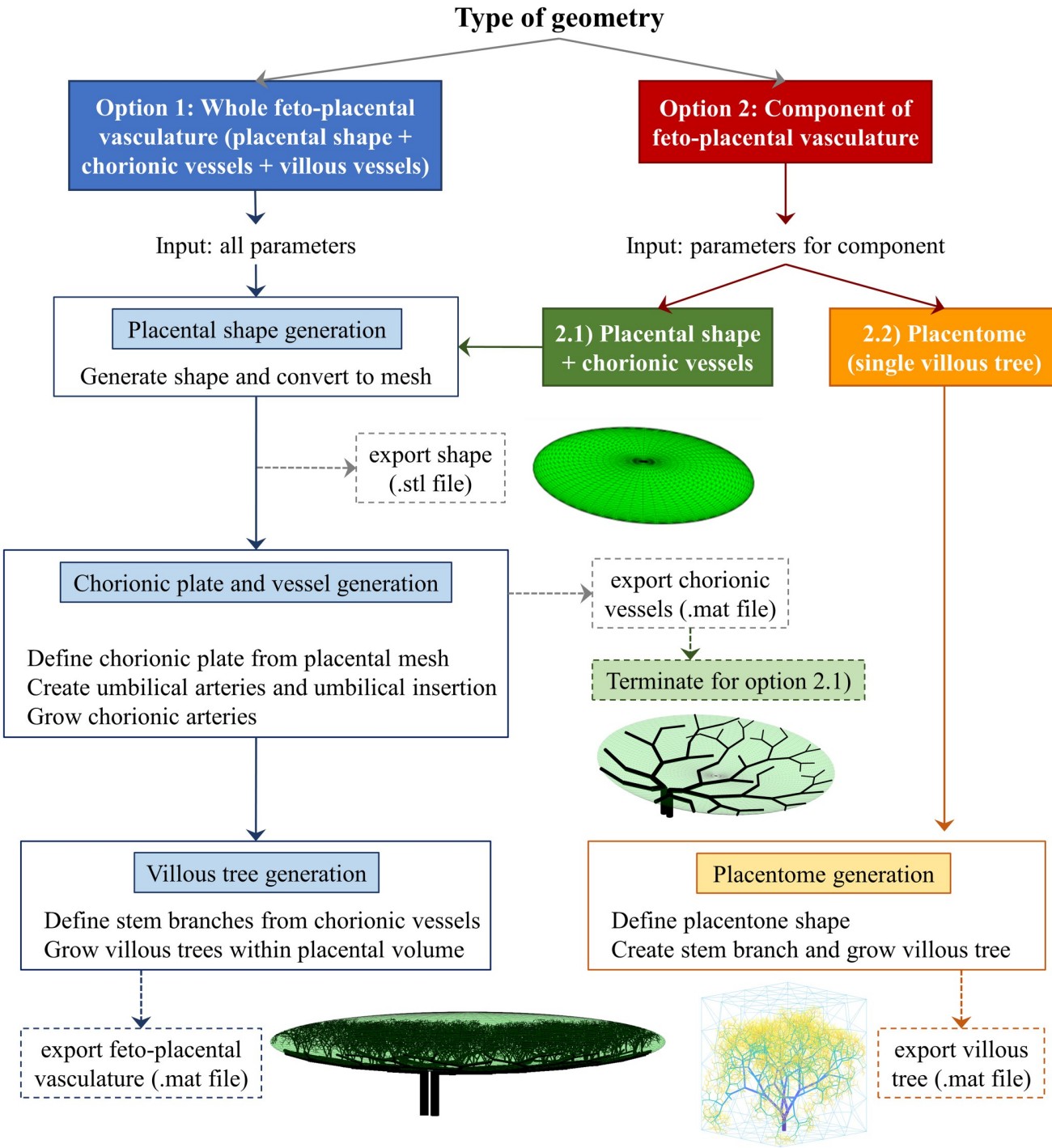

**Fig 3. User options and main steps of the generative algorithm.**

- General notions for vasculature generation in the placenta (General notions of feto-placental vasculature generation)
- The definition of chorionic plate and generation of chorionic vessels (Step-by-step description of chorionic vessel generation)

- The definition of a placentone domain (Step-by-step description of placentone domain generation) and the creation of villous trees (Step-by-step description of villous vessels generation)

- The representation of the capillary and venous systems (Capillary representation and venous system)

Depending on the end purpose, the algorithm can generate each part of the feto-placental vasculature independently or the full geometry. A flowchart of the algorithm options and an overview of the generative algorithm is provided in Fig 3.

**Tree notation.** The notation of the Trees Toolbox [45] is applied to all feto-vascular structures generated with the algorithm, specifically the chorionic vessels and the villous trees. We define a vascular tree by a series of start and end nodes (defined by coordinates in a 3D space), an adjacency matrix, and diameters. For a tree with $N$ nodes we have

$$\mathbf{C}_{1..N} = \{C_1, C_2, \ldots, C_N\}, \text{ where } C_i = (X_i, Y_i, Z_i) \in \mathbb{R}^3, \tag{3}$$

where $C_i$ is the set of all tree nodes defined by Cartesian coordinates $(X_i, Y_i, Z_i)$. The connections between nodes—or vessel segments—are denoted by the adjacency matrix $\mathbf{A}$, an $N \times N$ matrix with entries

$$A_{ij} = \begin{cases} 1, & \text{if node } i \text{ has parent } j \\ 0, & \text{otherwise.} \end{cases} \tag{4}$$

These connections yield a tree with $M$ segments defined as

$$\mathbf{B}_{1..M} = \{B_1, B_2, \ldots, B_M\}, \tag{5}$$

where each segment is characterised by a certain diameter $d$

$$d_{1..M} = \{d_1, d_2, \ldots, d_M\}, \tag{6}$$

where $d_i$ denotes the diameter of the connection between node $i$ and its parent node. This definition of a vascular tree resembles that of previous computational approaches [20, 32, 46].

**Branching rules.** Assuming that all segments are circular cross-section cylinders, the bifurcation of a parent segment into two daughter segments is dictated by Murray's Law, a power-law relationship between the diameters of the parent vessel $d_0$ and those of the daughter segments $d_1, d_2$:

$$d_0^k = d_1^k + d_2^k, \tag{7}$$

where $d_0 > d_1 = d_2$ for a symmetric bifurcation. This law minimises the resistance to flow throughout a vascular network, and the bifurcation exponent $k$ is key to determine the energy dissipation associated with a certain vascular structure.

## Step-by-step description of placental size and shape

**Motivation.** Post-partum [35, 37] and *in vivo* MRI [47] studies assessing placental shape found common deviations from a mean round shape. This can be quantified through *placental eccentricity*, a parameter that mathematically ranges between 0 and 1 (0 = completely circular shape), although typical *placental eccentricities* are well below 1 [10, 36, 37].

**Generative procedure.** We characterise our *in silico* placentas as oblate spheroids based on user-defined parameters:

$V$ : placental volume

$r_{maj}$ : long placental radius

$E$ : placental eccentricity

The placental thickness $2t_{half}$ is estimated from the placental volume, $V$, using Eq 8 [48, 49]:

$$t_{half} = \frac{3V}{4\pi r_{maj}^2}.$$
(8)

We use the equation of Pathak *et al.* [36] to obtain the corresponding short placental radius ($r_{min}$) from the *placental eccentricity* ($E$) and long placental radius ($r_{maj}$):

$$r_{min} = \sqrt{r_{maj}^2(1 - E^2)}.$$
(9)

For given values of $r_{maj}$, $r_{min}$ and $2t_{half}$, we mathematically define a 3D oblate spheroid as a surface object, characterised by 3D spatial coordinates. We employ a default patch conversion function in MATLAB to convert the geometry from this surface object to a patch structure containing directly associated face and vertex information [50]. This is represented as a closed quadrangular surface mesh whose elements are then further subdivided to form triangular elements, yielding a triangulated surface mesh. This mesh will be used to provide seed points for the vasculature growing algorithms (more details in subsequent sections). Alternatively, a patient-specific closed triangulated mesh can be directly obtained from *in vivo* imaging, such as placental MRI.

## Step-by-step description of placental mesh and cord insertion

**Motivation.** The umbilical cord inserts into the chorionic plate and gives rise to vessels spreading over the chorionic plate [51]. A measure of relative insertion eccentricity of the umbilical cord has been proposed by Pathak *et al.* [36]: the umbilical cord centrality index. This parameter ranges between 0 (insertion at plate centroid) and 1 (insertion at plate margin) [36].

**Generative procedure.** We characterise the shape of the chorionic plate and cord insertion based on two main user-defined parameters:

$CCI$ : umbilical cord centrality index

$mt$ : minimum distance between cord insertion point and chorionic plate periphery

The placental mesh is divided into chorionic and basal plates. First, $r_{maj}$, $r_{min}$ and $t_{half}$ are used to estimate the chorionic surface area, $A_{chor}$, based on the equation for half of the area of an ellipsoid:

$$A_{chor} = 2\pi \left( \frac{(r_{maj}r_{min})^{1.6} + (r_{maj}t_{half})^{1.6} + (r_{min}t_{half})^{1.6}}{3} \right)^{1/1.6}.$$
(10)

This surface area is then used to extract the mesh corresponding to the chorionic plate from the total placental mesh. This is performed by iteratively removing rows of mesh elements until the actual chorionic mesh surface area matches the area obtained using Eq 10. We found that the resulting mesh density had insufficient seed points and therefore hindered chorionic vessel generation (S1 Table). As such, we performed sub-triangulation of mesh elements to

provide sufficient seed points for the generative process, yielding 13—30 seeds/mm$^2$ (depending on the size of the placental surface). We use the equation of Pathak *et al.* [36] to determine the umbilical vessel insertion in the chorionic plate mesh. Specifically, we employ the umbilical cord centrality index, *CCI*, to determine the distance of the umbilical cord from the chorionic plate centroid, $d_c$:

$$d_c = CCI \cdot r_{maj}. \tag{11}$$

A search is then performed on the chorionic plate mesh nodes to find the node which matches $d_c$ the closest. This node is then assumed as the 3D insertion point in the chorionic plate mesh.

## General notions of feto-placental vasculature generation

Both chorionic and villous vessels are characterised by a range of user-defined parameters and, despite being based on different morphological data and being generated on different domains (chorionic—2D mesh surface; villous—3D domain), the basics for their generative procedure are similar. Broadly speaking, they are created iteratively, branching generation after branching generation, following four main steps:

1. Sampling of candidate daughter nodes based on morphological (e.g. branching angles, length-to-diameter ratios) and branching law (bifurcation exponents, asymmetry values) parameters;

2. Selecting the best candidate daughter nodes based on global branch distribution penalties, which are used to influence the spatial distribution of vessels in the chorionic plate and the IVS (more details in Methods—Computational implementation: Fine-tuning spatial distribution of feto-placental vasculature);

3. Ensuring there are no branch intersections;

4. Assessing criteria for termination at each iteration.

To facilitate the selection of candidate daughter nodes, heuristic tolerances in branch lengths determined from *ltd* ($tol_l$) and parent-daughter branching angles ($tol_\theta$) are allowed. Final branch lengths ($l_f$) and branching angles ($\theta_f$) then take the form

$$l_f = l \pm tol_l \quad \text{and} \quad \theta_f = \theta_{pd} \pm tol_\theta. \tag{12}$$

Users have the flexibility to define the tolerances $tol_l$ and $tol_\theta$; For all experiments in this paper, we heuristically selected $tol_l$ and $tol_\theta$ to align with user-defined constraints while ensuring the creation of a vascular structure containing an appropriate number of vessels, consistent with prior *in silico* models and observations from *in vivo/ex vivo* studies. We found that reducing the tolerances resulted in fewer candidate daughter nodes being selected and fewer vessels being generated, leading to an abrupt termination of vascular structures in both chorionic and villous vessels (see S3 and S6 Tables). Specifically, lower values of $tol_\theta$ hindered appropriate spread of chorionic vessels through the chorionic plate (see S4 Table). Ultimately, we defined $tol_l$ = 8–13% and $tol_\theta$ = 25–35% for generating chorionic vessels, while for villous vessels we set $tol_\theta$ = 15%. Users wishing to generate significantly different feto-placental vascular structures may need to modify these tolerances accordingly.

## Step-by-step description of chorionic vessel generation

**Motivation.** Chorionic vessels typically branch 6–8 times over the chorionic plate [51], which, in a healthy scenario, will then feed 30–100 villous trees in the IVS [3, 43, 52].

According to literature descriptions, the chorionic arteries have a mostly dichotomous branching pattern, with average parent-daughter diameter ratios ranging between 0.76 to 0.8 [51].

**Generative procedure.**   We characterise the generation of chorionic vessels based on user-defined parameters:

$ud$  : umbilical artery diameter

$k_c$  : chorionic Murrays Law bifurcation exponent

$\theta_{p_d}$  : parent-daughter branching angle

$\theta_{dd}$  : daughter-daughter branching angle

$ltd_c$  : chorionic length-to-diameter ratio

$a$  : asymmetry of branching generations

$bg_c$  : maximum number of branching generations

$bn_c$  : maximum number of segments

$cf_1$  : global distribution penalty weight 1 (distance to other vascular tree nodes)

$cf_2$  : global distribution penalty weight 2 (distance to chorionic plate centroid)

The distal ends of the umbilical arteries are represented by two segments 20 mm long [20]. These segments are inserted into the chorionic plate close to the designated point of insertion, ensuring no intersections by displacing their position in $x$ by a distance = $ud$. Chorionic arteries are generated iteratively from these cord insertions, spreading through the chorionic mesh. For each non-terminal end node, candidate branching nodes are selected from the chorionic mesh based on the desired daughter segment length and branching length priors within predefined tolerances (Eq 12). These nodes are further selected and ranked using a global distribution penalty. They are iteratively tested for intersection and termination criteria following the determined rank and a node is accepted if it respects the criteria. The subsequent creation of villous trees in the placenta involves creating intraplacental vessels of 2 mm in length. These originate from the midpoint of each chorionic artery and penetrate the IVS at angles ranging from 60 to 90 degrees [51].

**Branch termination.**   Biologically-inspired criteria: Based on previous studies [43, 51, 52], branching occurs until either the maximum number of branching generations ($bg_c$) is 8, or the maximum number of vessel segments ($bn_c$) is 100.

Topological criteria: We compute the Euclidean distance of a daughter branch node 1) to other tree points and 2) to the nodes in the chorionic plate boundary. If this distance is less than the daughter branch diameter in the first case or less than two times the daughter branch diameter in the second case, the branch is terminated. This ensures appropriate distance between vessel segments.

## Step-by-step description of placentone domain generation

**Motivation.**   Villous trees are associated with high variability in vascular density within the IVS [53]. However, they have been described as hollow-centred, bud-like structures supplied by a maternal SA located near their centres [3, 43]. Such functional units are commonly described as placentones, which have been characterised as semi-hemispheres [5] or cuboids [3].

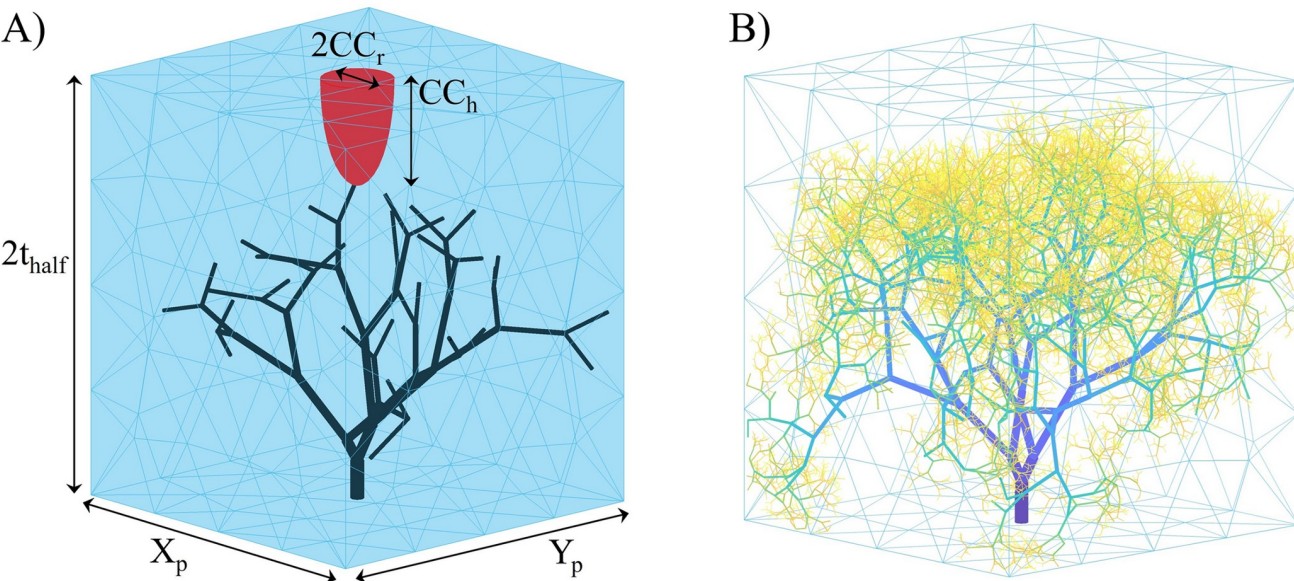

**Fig 4. Placentone representation.** (A) Main placentone dimensions are identified, including those for the central cavity, and a fetal tree with five branching generations is showcased for clarity. (B) A fetal tree branched up to 14 generations is displayed. Key: $2t_{half}$, placental thickness; $X_p$, x-direction length; $Y_p$, y-direction length; $CC_r$, central cavity radius; $CC_h$, central cavity height.

**Generative procedure.** To generate a placentone surface, the following parameters are required

$$
\begin{aligned}
V &\quad : \text{placental volume} \\
2t_{half} &\quad : \text{placental thickness} \\
n_p &\quad : \text{number of placentones} \\
CC_r &\quad : \text{central cavity radius} \\
CC_h &\quad : \text{central cavity height}
\end{aligned}
$$

To create a placentone, a cuboidal closed triangular mesh is initially created. This method assumes that the placenta can be approximated as uniform in width over a single placentone. The placentone, represented in Fig 4A, is assumed to have thickness equal to that of the placenta ($2t_{half}$) and cross-sectional dimensions $X_p = Y_p$ [3]. Cross-sectional dimensions are defined via Eq 13:

$$
X_p = Y_p = \sqrt{\frac{V}{2t_{half} \cdot n_p}}
\tag{13}
$$

where $V$ and $n_p$ represent the placental volume and the number of placentones, respectively. Furthermore, a central cavity was delineated, positioned adjacent to the opening of the SA and characterised as a semi-ellipsoid. The dimensions of this villous-free area, denoted by its radius ($CC_r$) and height ($CC_h$), were used to define its shape. Fetal villi are not allowed to grow within this central cavity.

### Step-by-step description of villous vessels generation

**Motivation.** Villous vessels originate from intraplacental vessels (or stem vessels, see Section Step-by-step description of chorionic vessel generation) and define the IVS, branching up to 15 generations and forming tree-like geometries [20, 43].

**Generative procedure.** Villous trees grow inside a surface mesh domain which effectively represents a $3D$ volume. Their generation process varies depending on whether they are generated as a single functional unit (mesh domain = placentone surface), or as part of the whole feto-placental vasculature (mesh domain = placental surface). In the first case (Section Step-by-step description of placentone domain generation), supplementary parameters are selected to delineate (1) a cuboid in which the tree grows, and (2) a villous-free region near the SA opening (central cavity) [3].

The subsequent parameters defined by the user describe the formation of villous trees for any mesh domain:

$$
\begin{aligned}
d_s &: \text{stem diameter} \\
ltd_v &: \text{villous length-to-diameter ratio} \\
k_v &: \text{villous tree Murrays Law bifurcation exponent} \\
a &: \text{asymmetry of branching generations} \\
\theta_{pd} &: \text{parent-daughter branching angle} \\
\theta_{dd} &: \text{minimum daughter-daughter branching angle} \\
cf_1 &: \text{global distribution penalty weight 1 (distance to plane defined by } n_V) \\
cf_2 &: \text{global distribution penalty weight 2 (distance to basal plate)} \\
bg_v &: \text{maximum number of branching generations}
\end{aligned}
$$

The selection of candidate branching nodes and branch generation is schematised in Fig 5. For each non-terminal end node $C$, candidate branching nodes are uniformly created on a spherical triangulated surface of radius $l_i$, centered around the parent node [54]. A plane is then defined by three points ($C_g$, current node; $C_{g-1}$, parent node; $C_{g-2}$ parent-parent node), whose unit normal ($n_V$) is obtained via Eq 14:

$$n_V = ||(C_{g-1} - C_g) \times (C_{g-2} - C_g)||. \tag{14}$$

Candidate nodes are split using this plane and those respecting branching angle priors within predefined tolerances (Eq 12) are further selected and ranked using a global distribution penalty. The nodes are iteratively tested for intersection and termination criteria following the determined rank and a node is accepted if it respects the criteria. An example of a fetal tree with 14 branching generations is showcased in Fig 4B.

**Branch termination.** Biologically-inspired criteria: The caliber of villous arteries is assumed to progressively decrease at each branching generation to feed terminal vessels of $d \sim 0.03$–$0.04$ mm, as observed in previous computational studies [20, 21]. This is achieved after 13–15 branching generations ($bg_v$).

Topological criteria: We compute the Euclidean distance of a daughter branch node 1) to other tree points and 2) to the nodes in the basal plate. We assume that, if this distance is less than the daughter branch diameter in the first case or less than two times the daughter branch diameter in the first case, the branch is terminated.

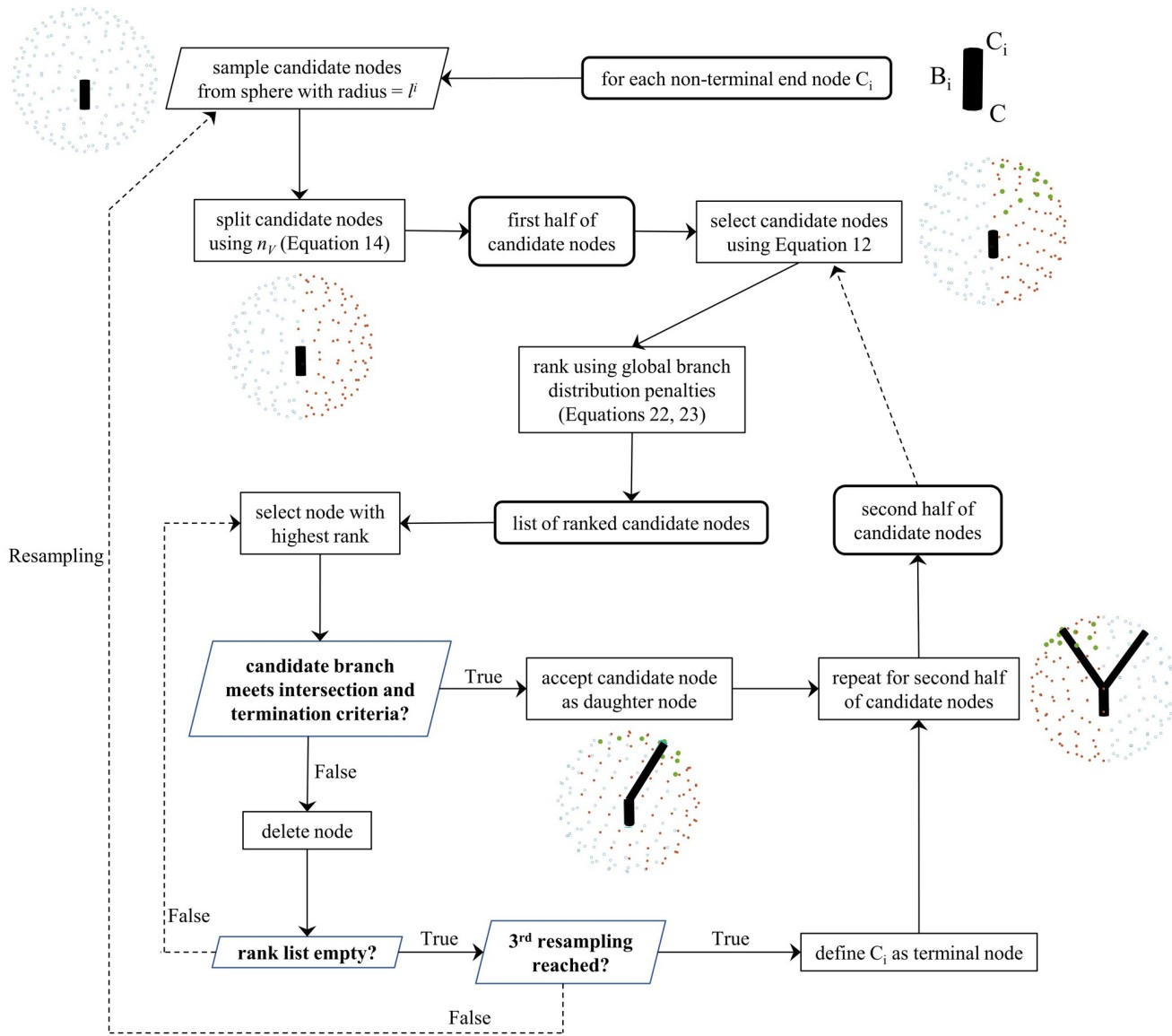

**Fig 5. Schematic showcasing main steps involved in the generation of a new daughter node $C_i$ and subsequently a new branch $B_i$.** Key: $l^i$, candidate branch length; $n_v$, unit normal of plane defined by Eq 14.

## Capillary representation and venous system

The current generative algorithm generates the arterial network of the feto-placental vasculature, from the umbilical cord level to intermediate villi, accounting for empty space to be occupied by the venous feto-placental counterpart (more details in Sections Computational implementation: Branch intersection checks and Morphological metrics computed). To create an additional representation of downstream vessels (i.e. mature intermediate villous vessels and capillaries), these can be grouped together into a lumped-parameter model, with a detailed description provided by Clark *et al.* [20].

### Analysis of algorithm outputs

**Morphological metrics computed.**   To assess the morphology of vascular structures generated with the algorithm, a range of topological properties were computed. These are established metrics for determining the functional behaviour of vascular networks [20, 21]. This includes properties directly obtained via functions from the MATLAB Trees Toolbox [45] and direct assessment from outputted structures, such as branching angles, daughter-to-mother diameter ratio, number of arterial branching generations, terminal vessel length and diameter, path lengths, number of segments, length-to-diameter ratios and maximum Strahler order. Other parameters were defined as follows:

**Strahler ratios.**   Branching metrics such as the Strahler branching ratio, diameter ratio and length ratio aim to quantify bifurcation characteristics within a vascular network. Different bifurcation properties are directly related to blood flow distributions, and differ between healthy and dysfunctional vascular structures [55]. The Strahler order of all branching nodes obtained with the MATLAB Trees Toolbox was used to compute the Strahler branching ratio, diameter ratio and length ratio. Each vessel segment is assigned with a Strahler order (obtained from the node Strahler order) and the number of segments in each order is counted. The number of segments, diameters and lengths are transformed into a logarithmic scale. Regression lines of these against the order number are determined via linear polynomial fitting, with each Strahler ratio being the antilog of the fit gradient [56–58].

**Vascular density.**   Mean vascular density is defined as the ratio between total vessel volume and the volume of the placenta (if growing in a complete placental structure) or the volume of the placentone (if growing as a unique fetal tree). We account for venous vascular density by including a volume term in calculations for total vessel volume: for a certain fetal tree branch $B_i$, venous length ($l_{vi}$) and diameter ($d_{vi}$) is expressed in function of arterial length ($l_{ai}$) and diameter ($d_{ai}$) [59],

$$l_{vi} = \frac{l_{ai}}{2} \quad \text{and} \quad d_{vi} = \frac{3}{2} \, d_{ai}, \tag{15}$$

and total vessel volume (v) is calculated as

$$v_t = v_a + v_v = \sum_{i=1}^{M} \pi \cdot \left(\frac{d_{ai}}{2}\right)^2 \cdot l_{ai} + \sum_{i=1}^{M} \pi \cdot \left(\frac{d_{vi}}{2}\right)^2 \cdot l_{vi}. \tag{16}$$

To compare these densities with micro-computed tomography estimates, we used a diameter cut-off equal to the voxel size achievable with their whole placental imaging ($\geq 116.5 \, \mu$m) and excluded vessels with diameter $\geq 0.9$ mm (equivalent to chorionic, not villous, vessels) [27].

**Vessel spread.**   Spread is defined as the standard deviation of the Euclidean distance of all vessels to the placental centroid $G_i$. Remembering that a tree has $N$ nodes defined by $C_i = (X_i, Y_i, Z_i)$, and assuming a list of candidate daughter nodes defined by $\delta_i = (X_i, Y_i, Z_i)$, we define

$$\text{spread} = \sqrt{\frac{\sum_{i=1}^{3} (\delta_i - G_i)^2}{n}} \tag{17}$$

where $n$ is the number of vessels.

**Topological assessment and sensitivity analysis.**   A global sensitivity analysis to assess the effect of algorithm input parameters on key topological metrics was performed for the generation of chorionic vessels and a fetal tree in a placentone. We employed Monte Carlo simulations with the Morris method, with this method known for its accuracy in models with many parameters and based on the repetition of a set of randomized one-at-a-time runs [60, 61].

This allows us to understand the relative importance of each input variable and how it affects the model's output. This is particularly useful for identifying key factors that significantly influence the results. A detailed description on the Morris method can be found elsewhere [61].

Input parameters assessed for chorionic and villous vessel generation are presented in Table 1. These ranges span biologically-plausible settings, accounting for extreme dysfunctional cases. The ranges reported for global distribution penalty weights 1 and 2 ($cf_1$, $cf_2$) were determined heuristically by investigating how adjustments to these weights impacted 1) the generation and spread of chorionic vessels across the chorionic plate and 2) the creation of a single fetal vascular structure. The placental outer shape, chorionic plate and umbilical cord insertion were fixed in the case of chorionic vessel generation, with only vessel growth input parameters being assessed. A uniform distribution of input parameters was imposed, with 70 and 100 samples per parameter for chorionic and villous vessel generation, respectively. A Latin hypercube sampling strategy was adopted to generate a near-random sample of parameter values using the uniform distribution previously defined [60]. This type of sampling splits the input parameter space into equally probable intervals along each input parameter while ensuring that the combinations of values chosen for the variables cover the entire input space in a representative manner.

After all runs, elementary effects (EE) associated with a certain input were computed as an absolute mean ($\mu^*$) and standard deviation ($\sigma$), which measure input influence and level of interactions with other inputs, respectively [60, 61]. Output parameters assessed for both chorionic

**Table 1. Sensitivity analysis settings.** Range of user-defined parameters for the generation of chorionic vessels and villous tree vessels.

| | Input | Ref. | Range |
|---|---|---|---|
| **Chorionic vessels** | $ud$ mean (mm) | [62] | 3.7–5.5 |
| | $ud$ SD (mm) | [62] | 0.6–1 |
| | $k_c$ | [63] | 3–3.4 |
| | $\theta_{pd}$(°) min | - | 0–20 |
| | $\theta_{pd}$(°) max | - | 20–80 |
| | $\theta_{dd}$(°) min | [51] | 50–90 |
| | $\theta_{dd}$(°) max | [51] | 90–130 |
| | $ltd_c$ SD | - | 0.4–1.5 |
| | $cf_1$ | - | 0.4–0.8 |
| | $cf_2$ | - | 0.1–0.5 |
| **Villous vessels** | $V$ (cm³) | [39, 48, 64] | 400–500 |
| | $2t_{half}$ (mm) | [49, 65] | 20–25 |
| | $d_s$ mean (mm) | [43, 51] | 0.51–0.89 |
| | $d_s$ SD (mm) | - | 0–0.1 |
| | $ltd_v$ mean | [20, 21] | 8–13 |
| | $ltd_v$ SD | [20, 21] | 0.5–5 |
| | $k_v$ | [63] | 2.5–3.5 |
| | $\theta_{pd}$(°) mean | [10, 20, 21] | 30–70 |
| | $\theta_{pd}$(°) SD | [10, 20, 21] | 0–30 |
| | $\theta_{dd}$(°) min | - | 15–40 |
| | $CC_r$ (mm) | [3] | 1–2.75 |
| | $CC_h$ (mm) | [3] | 3–13 |
| | $cf_1$ | - | 0.4–0.8 |
| | $cf_2$ | - | 0.2–0.6 |

**Table 2. Analysis of stochastic effects.** Input settings for the generation of chorionic vessels and villous tree vessels.

| Chorionic vessels | | Villous vessels | | |
|---|---|---|---|---|
| Input | Setting | Input | Healthy | FGR |
| $V$ (cm$^3$) | Unif(400, 500) | $V$ (cm$^3$) | 480 | 292.5 |
| $r_{maj}$ (cm) | $\mathcal{N}(9.3, 0.455)$ | $2t_{half}$ (mm) | 24 | 18 |
| $E$ | $\mathcal{N}(0.49, 0.17)$ | $n_p$ | 65 | 42 |
| $CCI$ | $\mathcal{N}(0.36, 0.21)$ | $CC_r$ (mm) | 1.88 | 1.88 |
| $mt$ (mm) | 20 | $CC_h$ (mm) | 5.6 | 5.6 |
| $ud$ (mm) | $\mathcal{N}(4.6, 0.9)$ | $d_s$ (mm) | $\mathcal{N}(0.7, 0.03)$ | $\mathcal{N}(0.56, 0.03)$ |
| $ltd_c$ | $\mathcal{N}(ltd_c(g), 1)$, Eq 27 | $ltd_v$ | $\mathcal{N}(10.5, 1)$ | $\mathcal{N}(10.5, 1)$ |
| $k_c$ | 3.2 | $k_v$ | 3.2 | 2.8 |
| $\theta_{dd}(°)$ | Unif(70, 100) | $a$ | $\mathcal{N}(1, 0.05)$ | $\mathcal{N}(1, 0.05)$ |
| $\theta_{pd}(°)$ | Unif(35, 50) | $\theta_{pd}(°)$ | $\mathcal{N}(45, 15)$ | $\mathcal{N}(54, 15)$ |
| $a$ | $\mathcal{N}(1, 0.05)$ | $\theta_{dd}(°)$ | 25 | 15 |
| $bg_c$ | 8 | $bg_v$ | 15 | 15 |
| $bn_c$ | 100 | $cf_1$ | 0.7 | 0.7 |
| $cf_1$ | 0.8 | $cf_2$ | 0.3 | 0.3 |
| $cf_2$ | 0.2 | - | - | - |

and villous vessels include: mean, minimum and maximum branching angle, mean number of branching generations, Strahler branching ratio, mean ratio of parent-to-daughter diameter and mean path length. Additional output parameters include vessel spread (Eq 17) for chorionic vessels and mean terminal diameter and vascular density for villous vessels, respectively.

To objectively determine highly influential input parameters, we normalised $\mu^*$ and $\sigma$ obtained for each input parameter and ranked these based on a cost function that maximises $\mu^*$ and minimises $\sigma$:

$$\text{score} = 0.5 \cdot \mu^* + 0.5 \cdot (1 - \sigma). \qquad (18)$$

The input parameters associated with the highest three scores as computed using Eq 18 were selected as the most influential for each output parameter.

**Stochastic effects.** To provide further insight into the variability inherent in the generated feto-placental vasculatures, multiple algorithm runs were conducted for the generation of chorionic vessels and placentone villous trees using the same set of input parameters. The input settings presented in Table 2 were used to run the algorithm 50 times for each case. The output metrics are well represented by the univariate Gaussian distributions. Variability between runs was quantified for each output metric by computing the Kullback-Leibler Divergence ($KL$) for the univariate Gaussians, two Gaussians at a time:

$$KL = \ln\left(\frac{\sigma_2}{\sigma_1}\right) + \frac{\sigma_1^2 + (\mu_1 - \mu_2)^2}{2 \cdot \sigma_2^2} - \frac{1}{2}, \qquad (19)$$

where $\mathcal{N}(\mu_1, \sigma_1)$ and $\mathcal{N}(\mu_2, \sigma_2)$ represent the mean and standard deviation values of Gaussians 1 and 2, respectively. $KL$ varies between 0 and $\infty$, and the lower its value, the closest the behaviour of two distributions.

**Generating healthy and dysfunctional feto-placental vasculatures.** To assess vascular spatial variability in the healthy placenta, we generated whole feto-placental vascular structures based on input parameters presented in Table 3. We also generated 3 placentones at 35 weeks of gestation based on input parameters presented in Table 4: a healthy fetal tree; the presence of

**Table 3. Input settings for the analysis of the whole feto-placental vasculature.** Analysis of vascular spatial variability of the complete feto-placental vasculature. Units: $V$, cm$^3$; $r_{maj}$, cm; $mt$, mm; $ud$, mm; $\theta_{pd}$ and $\theta_{dd}$, °; $d_s$, mm.

| Placental shape | | | | Umbilical insertion | | | | | |
|---|---|---|---|---|---|---|---|---|---|
| $V$ | $r_{maj}$ | $E$ | | $CCI$ | $mt$ | | | | |
| Unif(400, 500) | $\mathcal{N}(9.3, 0.455)$ | $\mathcal{N}(0.49, 0.17)$ | | $\mathcal{N}(0.36, 0.21)$ | 20 | | | | |
| Chorionic vessels | | | | | | | | | |
| $ud$ | $ltd_c$ | $k_c$ | $a$ | $\theta_{pd}$ | $\theta_{dd}$ | $cf_1$ | $cf_2$ | $bg_c$ | $bn_c$ |
| $\mathcal{N}(4.6, 0.9)$ | $\mathcal{N}(ltd_c(g), 1)$, Eq 27 | 3.2 | $\mathcal{N}(1, 0.05)$ | $\mathcal{N}(35, 50)$ | $\mathcal{N}(70, 100)$ | 0.8 | 0.2 | 8 | 100 |
| Villous vessels | | | | | | | | | |
| $d_s$ | $ltd_v$ | $k_v$ | $a$ | $\theta_{pd}$ | $\theta_{dd}$ | $cf_1$ | $cf_2$ | $bg_v$ | |
| $\mathcal{N}(0.7, 0.03)$ | $\mathcal{N}(10.5, 0.8)$ | 3.2 | $\mathcal{N}(1, 0.05)$ | $\mathcal{N}(45, 15)$ | 25 | 0.7 | 0.3 | 13 | |

a SA mega jet, associated with a higher central cavity height [3]; and moderate FGR, represented via a 35% decrease in placental volume [39, 66] and number of placentones [51], a 25% decrease in placental thickness $2t_{half}$ [44], a 20% smaller stem diameter [66], a 20% increase in mean parent-daughter branching angle [10] and a 13% decrease in the bifurcation exponent $k_v$.

## Results

In this section, we first present an example of an *in silico* feto-placental vasculature obtained with our generative algorithm and analyse its topological properties against literature ranges, for overall validation of the structure (Assessment of general topological properties and comparison with literature). This assessment is conducted by comparing predicted topological metrics against *ex vivo* and *in vivo* data whenever possible. *In silico* studies serve as the ultimate point of comparison when no other data is available. However, relying solely on comparisons between our topological predictions and other computational studies may be insufficient to validate the generated vascular structures; thus, these comparisons are presented merely as a benchmark. Next we assess the sensitivity analyses, evaluating the dependence of generated shapes on input parameters (Dependence of generated shape on input parameters). The stochastic effects of the algorithm are analysed in terms of key output metrics (Analysis of stochastic effects) and vascular spatial variability for two feto-placental vasculatures (Variability in vascular density: A two-case assessment). Finally, we perform a preliminary assessment of three different placentones representing healthy and dysfunctional scenarios (Placentone fetal trees).

### Assessment of general topological properties and comparison with literature

The feto-placental vasculature generated with input parameters from Table 3 is presented in Fig 6, with morphological characteristics and key output topological metrics presented in

**Table 4. Input settings for the analysis of three placentone cases.** A healthy fetal tree, a case with the presence of a spiral artery (SA) mega jet, and a case of fetal growth restriction (FGR) are assessed. Units: $V$, cm$^3$; $2t_{half}$, mm; $CC_r$, mm; $CC_h$, mm; $d_s$, mm; $\theta_{pd}$ and $\theta_{dd}$, °.

| Case | $V$ | $2t_{half}$ | $n_p$ | $CC_r$ | $CC_h$ | $d_s$ | $ltd_v$ | $k_v$ | $a$ | $\theta_{pd}$ | $\theta_{dd}$ | $cf_1$ | $cf_2$ | $bg_v$ |
|---|---|---|---|---|---|---|---|---|---|---|---|---|---|---|
| | | | | | | | | Input | | | | | | |
| Healthy | 390 | 24 | 65 | 1.88 | 5.6 | $\mathcal{N}(0.7, 0.03)$ | $\mathcal{N}(10.5, 0.8)$ | 3.2 | $\mathcal{N}(1, 0.05)$ | $\mathcal{N}(45, 15)$ | 25 | 0.7 | 0.3 | 15 |
| SA mega jet | 390 | 24 | 65 | 1.88 | 12.91 | $\mathcal{N}(0.7, 0.03)$ | $\mathcal{N}(10.5, 0.8)$ | 3.2 | $\mathcal{N}(1, 0.05)$ | $\mathcal{N}(45, 15)$ | 25 | 0.7 | 0.3 | 15 |
| FGR | 292.5 | 18 | 42 | 1.88 | 5.6 | $\mathcal{N}(0.56, 0.03)$ | $\mathcal{N}(10.5, 0.8)$ | 2.8 | $\mathcal{N}(1, 0.05)$ | $\mathcal{N}(54, 15)$ | 25 | 0.7 | 0.3 | 15 |

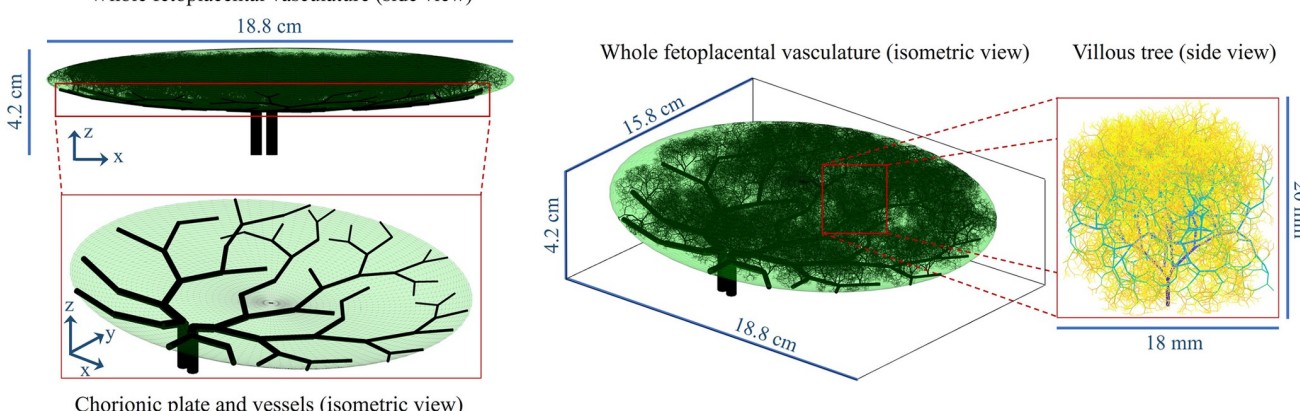

**Fig 6. Example of feto-placental vasculature generated in the case of a non-central umbilical cord insertion.** Figure components are depicted in isometric (*xyz* coordinate system) and side (*xz* plane) views.

Table 5. This structure includes up to 13 generations beyond the chorionic arteries, with a total of 21±4 mean branching generations up to terminal vessels of diameter 0.036±0.015 mm, consistent with previous reports [43, 51, 67]. Strahler branching properties in the generated vasculature show asymmetric vascular branching, with values comparable to those obtained by previous computational models of the feto-placental vasculature. Similarly, mean morphometric properties of the structure such as branching angles and length-to-diameter ratio correspond well to previous estimates for placental computational geometries accounting for heterogeneously generated fetal trees [21].

**Table 5. Generated feto-placental vasculature morphological properties and key topological metrics, obtained with main input parameters as described in previous sections.** Typical ranges of structural values available in the literature are also reported.

| Parameter | Literature range | Placental shape | Feto-placental vessels |
|---|---|---|---|
| $V$ (cm$^3$) | *in vivo*: 400–600 [39, 48, 64] | 409.1 | - |
| $r_{maj}$ (cm) | *in vivo*: $\sim \mathcal{N}(9.07, 0.181)$ [37, 49] | 8.47 | - |
| $E$ (mm) | *ex vivo*: $\sim \mathcal{N}(0.49, 0.17)$ [36] | 0.54 | - |
| $mt$ (mm) | other: 20 mm [68] | - | 20 |
| $ud$ (mm) | *in vivo*: $\sim \mathcal{N}(4.6, 0.9)$ [62] | - | 4.7 |
| $\theta_{pd}$ (°) mean ± SD | *ex vivo*: 40–70˚ [10, 69] | | |
| $\theta_{dd}$ (°) min | - | - | 13.28 |
| $\theta_{dd}$ (°) max | - | - | 90 |
| Mean length-to-diameter ratio ± SD | *in silico*: 8.46±5.63 [20, 21] | - | 10.93±1.26 |
| Mean daugther-to-mother diameter ratio ± SD | *ex vivo*: 0.66±0.15 [67]; *in silico*: 0.73±0.14 [20, 21]; *ex vivo*: 7.51 [5.48, 8.84] [59] | - | 0.79±0.006 |
| Maximum Strahler order | *in silico*: 11–12 [21] | - | 15 |
| Strahler branching ratio | *in silico*: 2.30–2.65 [20, 21]; *ex vivo*: 2.73 (chorionic vessels) [70] | - | 2.47 |
| Strahler diameter ratio | *in silico*: 1.50–1.53 [21] | - | 1.42 |
| Strahler length ratio | *in silico*: 1.23–1.41 [21] | - | 1.17 |
| Mean arterial branching generations (including chorionic vessels) ± SD | *ex vivo*: 16–23 [43, 51], 20.59±8.71 [67] | - | 21±4 |
| Mean diameter of terminal arterial segments ± SD | *ex vivo*: 0.03 [43]; *in silico*: 0.03–0.08 [20, 21] | - | 0.036±0.015 |

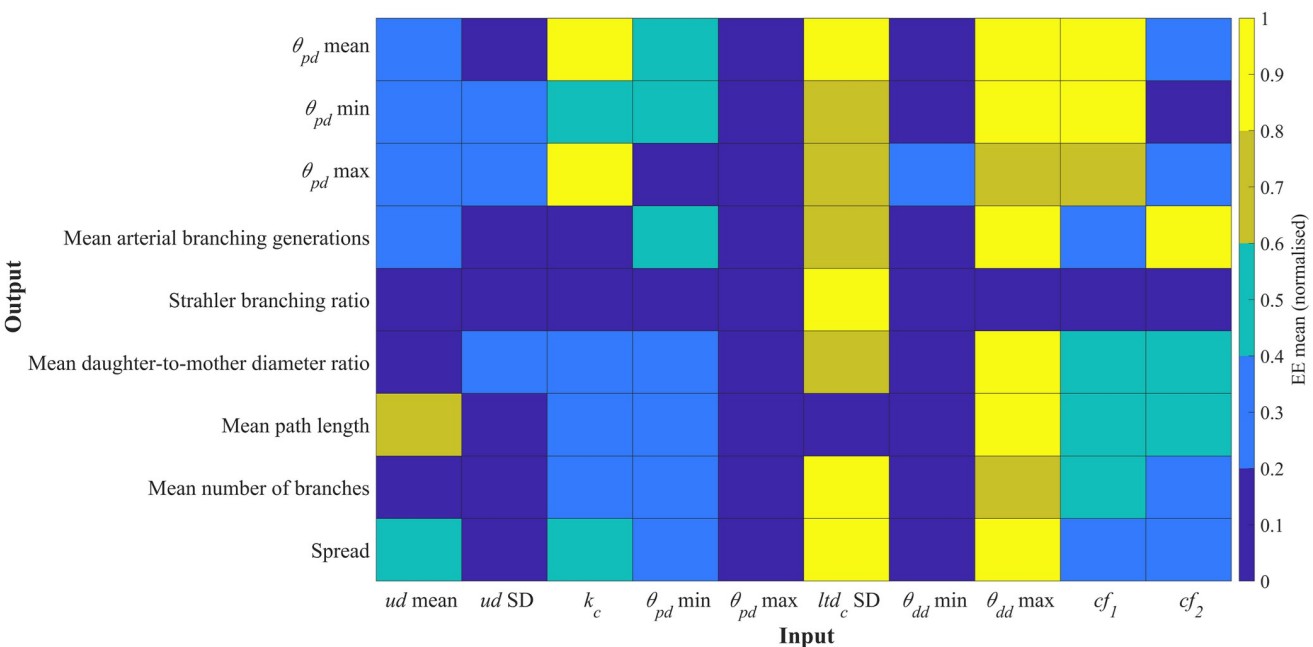

**Fig 7. Heatmap results from the global sensitivity analysis for the generation of chorionic vessels.** Mean ($\mu^*$) values of the elementary effects (EE) associated with different output metrics were obtained for an input space with 70 samples per input parameter (input settings from Table 1. $\mu^*$ are normalised for direct comparison between EEs, and higher $\mu^*$ values are associated with increased influence of a certain input over a certain output. Key: $ud$, umbilical artery diameter; $k_c$, chorionic Murray's Law bifurcation exponent; $\theta_{pd}$, parent-daughter branching angle; $ltd_c$, chorionic length-to-diameter ratio; $\theta_{dd}$, daughter-to-daughter branching angle; $cf_1$, $cf_2$, global distribution penalty weights.

## Dependence of generated shape on input parameters

The influence of input parameters on key topological metrics is assessed for chorionic (Chorionic vessels) and villous trees (Placentone). The relative changes in output metrics for chorionic vessels across a range of input parameters are shown in Fig 7, while the values of $\mu^*$ and $\sigma$ for the elementary effects (EE) of each input parameter are presented in Fig 8. Similarly, the corresponding results for placentone villous trees are displayed in Figs 9 and 10, respectively. The ratio $\sigma/\mu$, which allows us to characterise model parameters with regards to (non-)linearity, (non-) monotony or parameter interactions [71], is also plotted in Figs 8 and 10.

**Chorionic vessels.** Fig 7 indicates that the standard deviation of the chorionic length-to-diameter ratio ($ltd_c$) and the maximum daughter-daughter branching angle (max $\theta_{dd}$) significantly affect most output metrics. Mean, minimum, and maximum branching angles are sensitive to the bifurcation exponent $k_c$ and global distribution penalty weight $cf_1$. Interestingly, variations in input maximum $\theta_{pd}$ and minimum $\theta_{dd}$ seem to have no apparent impact on output metrics.

As displayed in Fig 8, every input parameter is significant since $\mu^* > 0$ for all parameters, and all exhibit a $\sigma/\mu$ ratio $>1$, indicating non-linear behavior, or interaction effects with other parameters, or a combination of both. In Fig 8, we have highlighted highly influential parameters obtained by maximising $\mu^*$ and minimising $\sigma$ (Eq 18). The findings presented align with the heatmap from Fig 7, emphasizing the crucial role of the $cf_1$ for all output metrics, and of the $cf_2$ and the maximum $\theta_{dd}$ for most output metrics. The mean number of arterial branching generations and Strahler branching ratios are also impacted by the minimum $\theta_{pd}$, while $k_c$ impacts the mean daughter-to-mother diameter ratio and the mean path length. Interestingly, the standard deviation of the umbilical artery diameter, maximum $\theta_{pd}$ and minimum $\theta_{dd}$

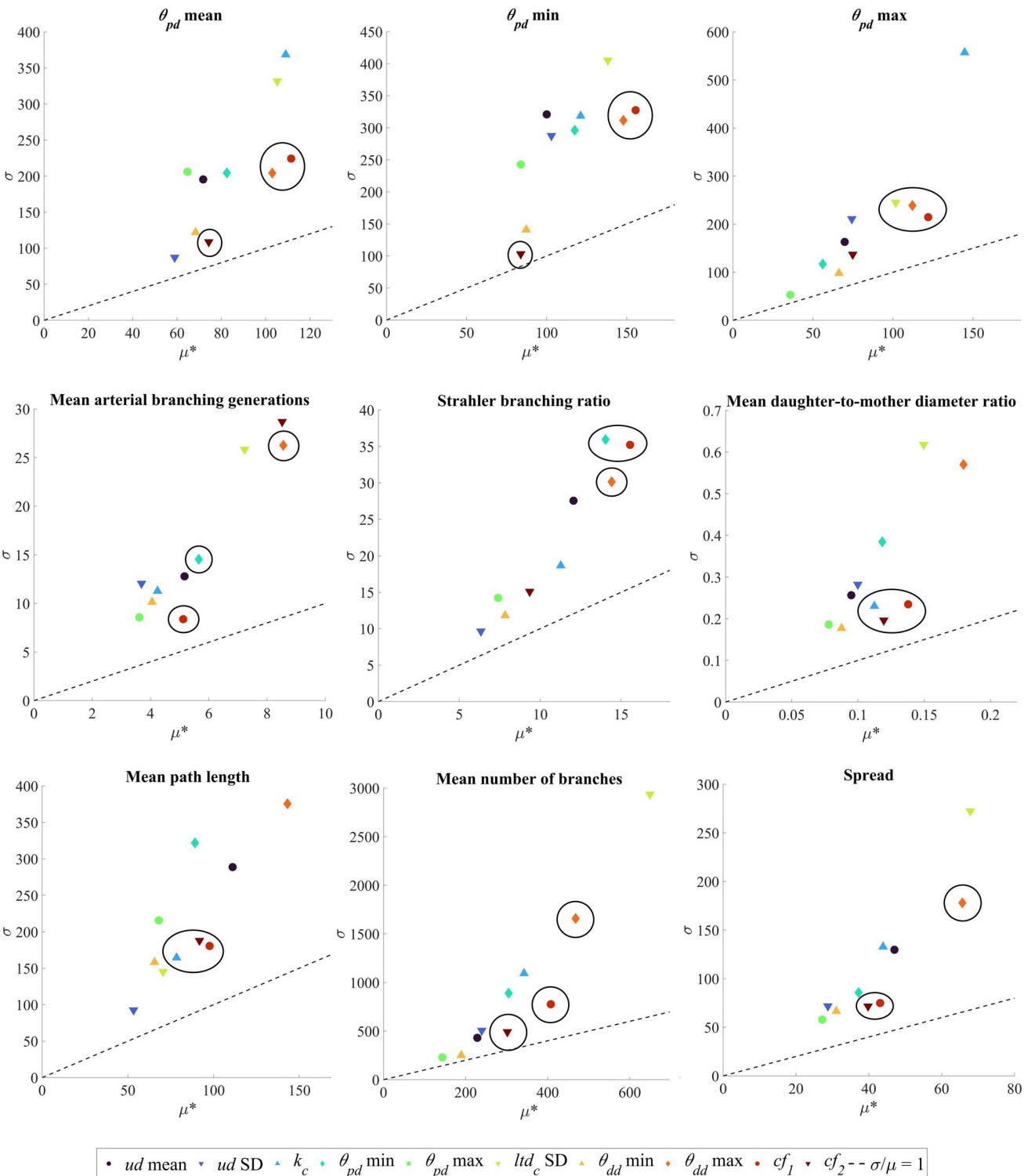

**Fig 8. Elementary effects (EE) results for chorionic vessel generation employing the input parameter ranges from Table 1.** Input parameters are listed and color-coded at the end of the Figure. Each subplot corresponds to one output metric indicated by subplot titles. For each output metric, the standard deviation ($\sigma$) associated with the EE of a certain input parameter is plotted in function of the respective mean ($\mu^*$). The $\sigma/\mu$ ratio is shown by dotted lines in all plots. Clusters of highly influential input parameters, as determined by Eq 18, are circled in black. Key: $ud$, umbilical artery diameter; $k_c$, chorionic Murray's Law bifurcation exponent; $\theta_{pd}$, parent-daughter branching angle; $ltd_c$, chorionic length-to-diameter ratio; $\theta_{dd}$, daughter-to-daughter branching angle; $cf_1$, $cf_2$, global distribution penalty weights.

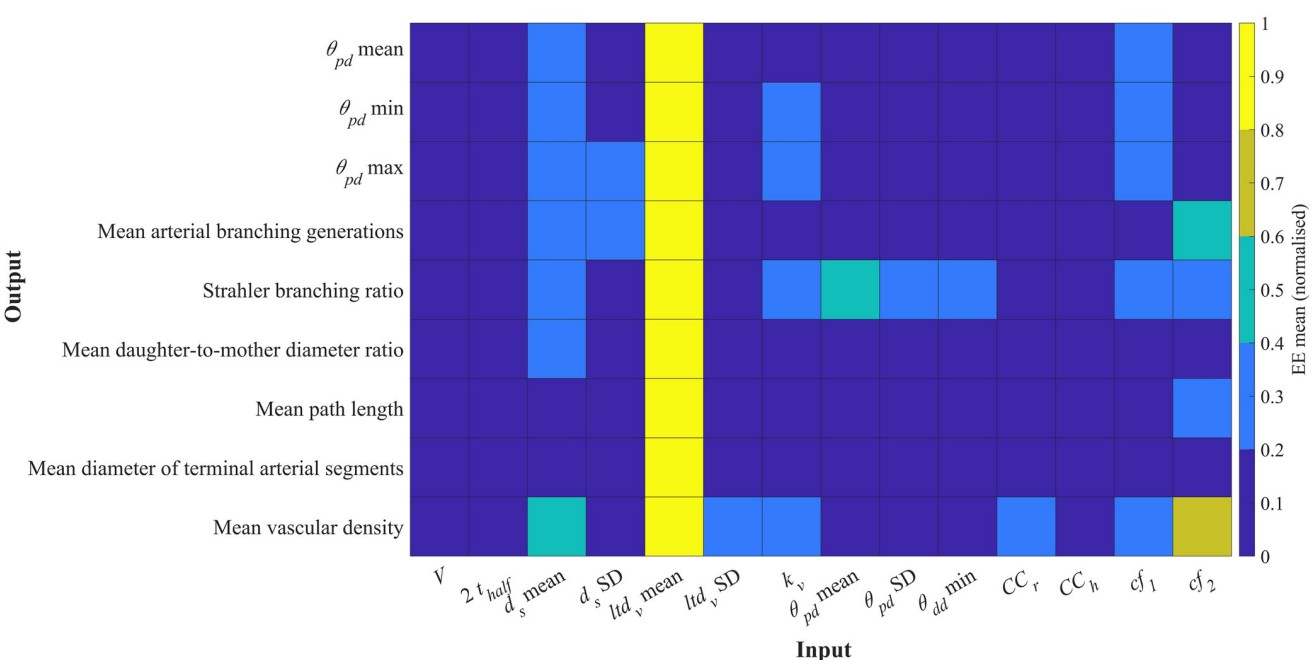

**Fig 9. Heatmap results from the global sensitivity analysis for the generation of villous vessels within a placentone.** Mean ($\mu^*$) values of the elementary effects (EE) associated with different output metrics were obtained for an input space with 100 samples per input parameter (input settings from Table 1). $\mu^*$ are normalised for direct comparison between EEs, and higher $\mu^*$ values are associated with increased influence of a certain input over a certain output. Key: $V$, placental volume; $2t_{half}$, placental thickness; $d_s$, stem diameter; $ltd_v$, villous length-to-diameter ratio; $k_v$, villous Murray's Law bifurcation exponent; $\theta_{pd}$, parent-daughter branching angle; $\theta_{dd}$, daughter-to-daughter branching angle; $CC_r$, central cavity radius; $CC_h$, central cavity height; $cf_1$, $cf_2$, global distribution penalty weights.

appear to have minimal influence on most output metrics. These results suggest that chorionic vessel generation is mainly influenced by maximum daughter-daughter branching angles, while the standard deviation of the umbilical artery diameter may not be essential in future algorithm setups.

**Placentone.** In Fig 9, we observe that the majority of input parameters have a moderate impact on the output metrics. The mean villous length-to-diameter ratio ($ltd_v$) appears to stand out as the most influential parameter, associated with the highest normalised EE mean, followed by the mean stem diameter ($d_s$). However, parameters related to the placentone's shape (e.g. placental volume—$V$—and thickness—$2t_{half}$) do not seem to significantly affect villous vessel topology. Additionally, it appears that only the standard deviation of the parent-daughter branching angle ($\theta_{pd}$) influences the Strahler branching ratio.

Our results from Fig 10 once again demonstrate non-linear behavior and parameter interactions, as indicated by $\sigma/\mu$ ratios >1. Despite the mean villous length-to-diameter ratio ($ltd_v$) standing out as the most influential parameter in Fig 9, its consistently high $\sigma$ across all output parameters, as depicted in Fig 10, hinders its classification as a highly influential parameter based on the ranking from Eq 18. Notably, the mean stem diameter ($d_s$) remains as a highly influential parameter, impacting all output metrics. The mean, minimum and maximum branching angles are influenced by the bifurcation exponent ($k_v$), mean parent-daughter branching angle ($\theta_{pd}$), minimum daughter-to-daughter branching angle ($\theta_{dd}$) and global distribution penalty weights ($cf_1$, $cf_2$). The bifurcation exponent also dictates other output metrics such as the Strahler branching ratio, the mean diameter of terminal arterial segments and mean vascular density. Various output metrics show low sensitivity to different input

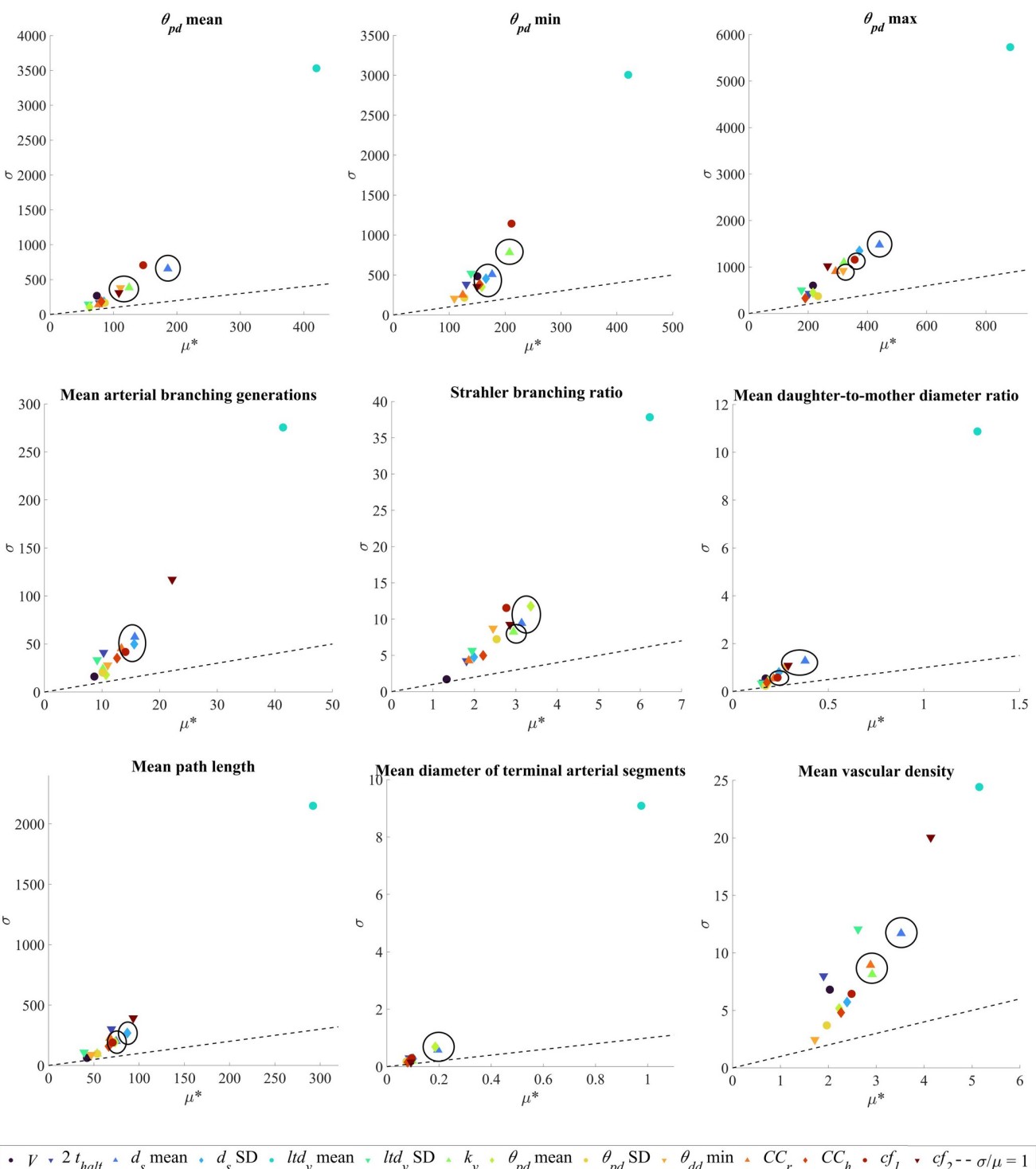

**Fig 10. Elementary effects (EE) results for the generation of villous vessels within a placentone employing the input parameter ranges from Table 1.**
Input parameters are listed and color-coded at the end of the Figure. Each subplot corresponds to one output metric indicated by subplot titles. For each output metric, the standard deviation ($\sigma$) associated with the EE of a certain input parameter is plotted in function of the respective mean ($\mu^*$). The $\sigma/\mu$ ratio is shown by dotted lines in all plots. Clusters of highly influential input parameters, as determined by Eq 18, are circled in black. Key: $V$, placental volume; $2t_{half}$, placental thickness; $d_s$, stem diameter; $ltd_v$, villous length-to-diameter ratio; $k_v$, villous Murray's Law bifurcation exponent; $\theta_{pd}$, parent-daughter branching angle; $\theta_{dd}$, daughter-to-daughter branching angle; $CC_r$, central cavity radius; $CC_h$, central cavity height; $cf_1$, $cf_2$, global distribution penalty weights.

parameters. For instance, the standard deviation of $ltd_v$ has minimal impact on most outputs, suggesting its limited importance in villous tree generation. Parameters related to placentome shape (e.g., $V$, $2t_{half}$, $CC_r$, $CC_h$) do not significantly affect global branching angles, branching ratios, or average path lengths. These inputs mainly influence vessel distribution within a volume (e.g. mean vascular density) and may lead to localized changes in topological statistics without exerting widespread global effects.

### Analysis of stochastic effects

**Variability in output metrics from multiple algorithm runs.**   Key output metrics evaluated for variability between algorithm runs include vessel path lengths, branching angles, terminal diameters and lengths, length-to-diameter ratio and branching generations. This variability is statistically represented in Fig 11 using boxplots of *KL* divergence results. *KL* divergences obtained for healthy chorionic and villous vessels are represented in Fig 11A and 11B, while Fig 11C and 11D focus on the differences in *KL* divergence linked to healthy and dysfunctional placentone fetal trees. Additional output metrics are included in the Supporting information (S1 Fig).

The generative algorithm yields vascular structures which are stochastic in nature. We can observe high variability in output metrics between algorithm runs: the median *KL* divergence fluctuates between 0 and 0.46 for different output metrics in Fig 11A and 11B, with the upper quartile peaking at 1.19 in the case of chorionic vessels. Maximum boxplot scores (excluding outliers) range from 2 to 2.65. We observe a high *KL* divergence dispersion and the presence of outliers ($KL > 3$) for most of the metrics associated with chorionic vessel structures (Fig 11A). While a series of outliers remains for the metrics associated with villous tree structures (Fig 11B), the highest *KL* divergence dispersion is observed for the path length.

We can observe that the stochastic *KL* divergence for the output metrics associated with FGR fetal trees (Fig 11C) is higher in comparison with that depicted in Fig 11B) for healthy fetal trees. This is especially true for the mean path length output metric, where the *KL* divergence upper quartile exceeds 2, whereas for the healthy fetal tree it is below 1 (as shown in Fig 11B). This suggests that it is not possible to generate a dysfunctional vascular structure when using input parameters associated with a healthy one. (Fig 11D) displays even higher *KL* divergences for output metrics from paired healthy and FGR fetal trees. For instance, the *KL* divergence lower quartiles associated with mean path length, mean diameter of terminal segments and mean terminal length exceeds 6, while the upper quartile is beyond 20 for mean path length. These results further emphasize the topological differences between healthy and dysfunctional fetal vasculatures.

**Variability in vascular density: A two-case assessment.**   Two feto-placental vasculatures generated with input parameter settings from Section Generating healthy and dysfunctional feto-placental vasculatures and with different umbilical cord insertion locations were assessed for differences in vascular spatial variability, vascular density and supply asymmetry.

As displayed in Fig 12A and 12B, the feto-placental chorionic vasculatures generated include dichotomous and monopodial branching patterns. The percentage of monopodial branches in the non-central and centralised umbilical insertion vascular structures showcased here is 21% and 25%, respectively. This demonstrates that there is no preference for either branching mode with varying umbilical insertions. There is, however, asymmetry in the vascular volumes supplied by each umbilical artery, which is higher in the case of a non-central cord insertion (Fig 12A). For this structure, the arterial volume is distributed in a 0.37:0.63 ratio, while the vascular structure with a centralised cord insertion (Fig 12B) has a ratio of 0.48:0.52. In addition to this asymmetry in blood supply, these vascular structures have different mean

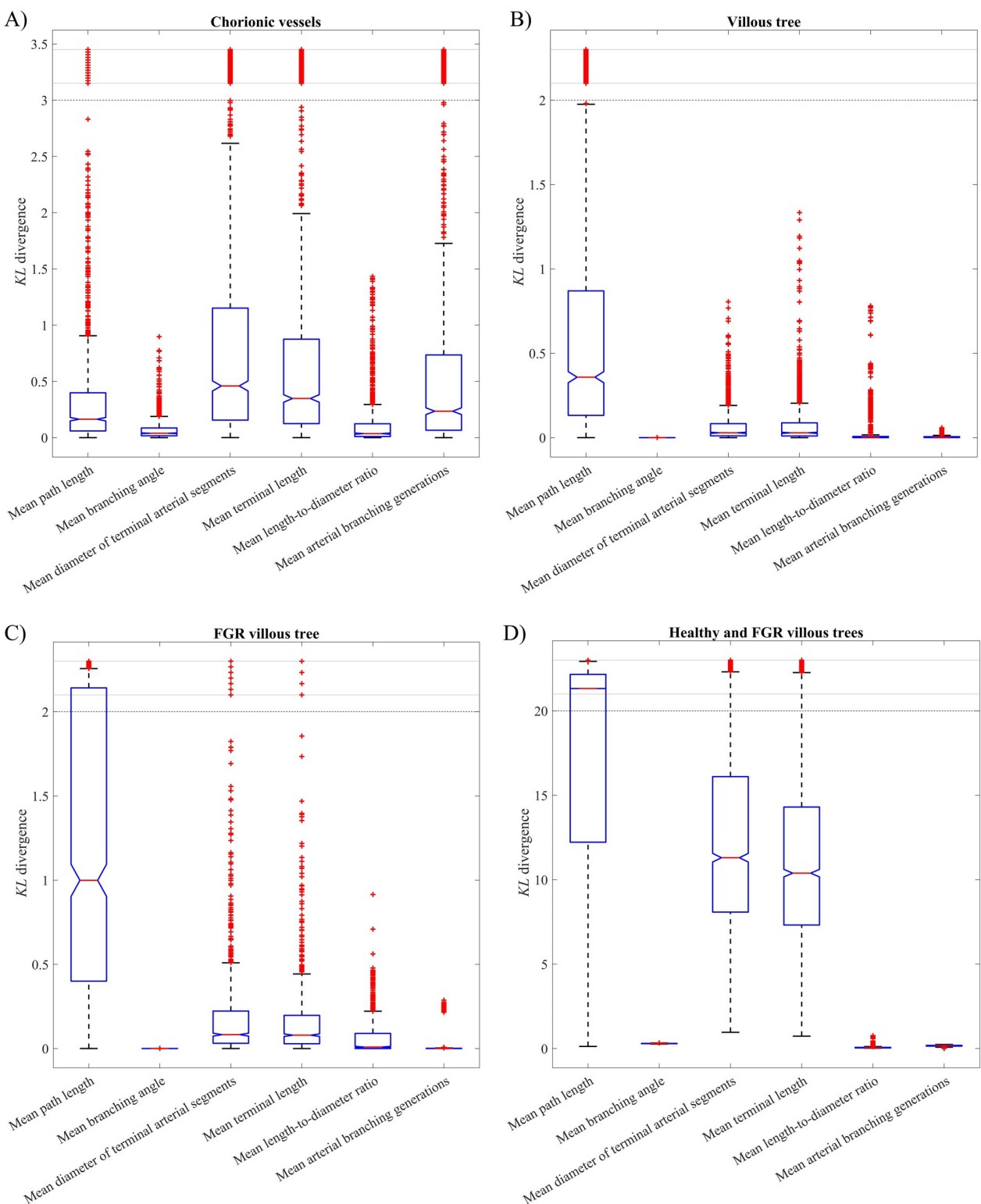

**Fig 11. Boxplots displaying *KL* divergence results, indicating variability of key topological metrics obtained after multiple algorithm runs for the generation of chorionic vessels (A) and villous vessels (B-D).** Each output metric is modeled as a Gaussian distribution, and Eq 24 calculates the *KL* divergence between pairs of Gaussians from the output space. With 50 algorithm runs, this results in 1225 Gaussian combinations and *KL* divergence values per output metric, represented for (A) chorionic vessels, (B) healthy villous vessels, and (C) FGR-associated villous vessels. (D) A similar analysis using 50 healthy and 50 FGR fetal trees was conducted, focusing on healthy-dysfunctional output metric pairs to derive the *KL* divergence.

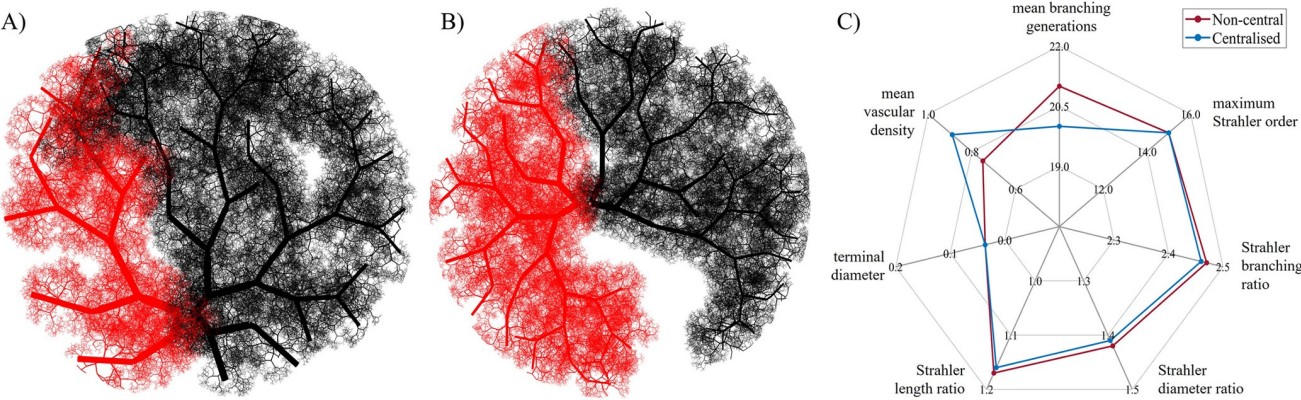

**Fig 12. Feto-placental vasculatures with non-central (A) and centralised (B) umbilical artery insertion.** Red (left vasculature) and black (right vasculature) represent vessels fed by each umbilical artery. A comparison of mean key vessel topological metrics is also displayed (C).

vascular densities (non-central: 0.75±0.32%; centralised: 0.89±0.38%); however, other key topological metrics for the whole feto-placental vascular structure remain mainly unaffected (Fig 12C).

Regarding the chorionic vasculature exclusively, additional topological metrics can demonstrate structural differences between centralised and non-central umbilical insertions. A centralised insertion is associated with 3.85±1.71 mean branching generations, spanning 3–5 generations to reach the placental periphery. On the other hand, a non-central insertion is associated with 4.27±2.04 mean branching generations, requiring 4–7 generations to reach the placental periphery. This difference in branching generations is also reflected in the entire pathway from the umbilical cord to terminal vessels, which is higher in the case of non-central insertion (125.09±28.84 mm) when compared to centralised insertion (121.17±14.15 mm). This is associated with increased variability in vessel pathways for a non-central insertion vasculature, as indicated by the increased standard deviation. Moreover, the vasculature with centralised insertion has a higher spread of chorionic vessels throughout the chorionic plate when compared to the one with non-central insertion (66.9 versus 58.4, respectively).

Fig 13 shows the spatial distribution of feto-placental vessels for each vascular structure, including (a) isosurface maps of nodal density and (b) mean vascular density maps averaged in the *z*-direction. No spatially consistent difference in villous vessel density can be observed within the placental volume, with both structures having a marked degree of heterogeneity. This is demonstrated by comparable coefficients of variation in vascular density for both structures (81.3% vs 82.1%). Both vasculatures are also characterised by reduced vascular density in the placental periphery, with higher vessel density inward.

## Placentone fetal trees

Key topological metrics for the three placentones created using input parameters from Section Generating healthy and dysfunctional feto-placental vasculatures are presented in Fig 14. Strahler diameter and length ratios remained relatively unaltered across placentones, with the FGR case yielding a higher maximum Strahler order and a smaller Strahler branching ratio in comparison with other placentones (2.24 vs 2.43 and 2.47), associated with more symmetric branching. Mean path lengths from stem trunks to terminal endpoints, also showcases in the first row of Fig 15, were similar for healthy and mega jet cases (26.80±1.87 and 26.40±2.18 mm). The presence of a SA mega jet leads to a spatial redistribution of villous vessels where the

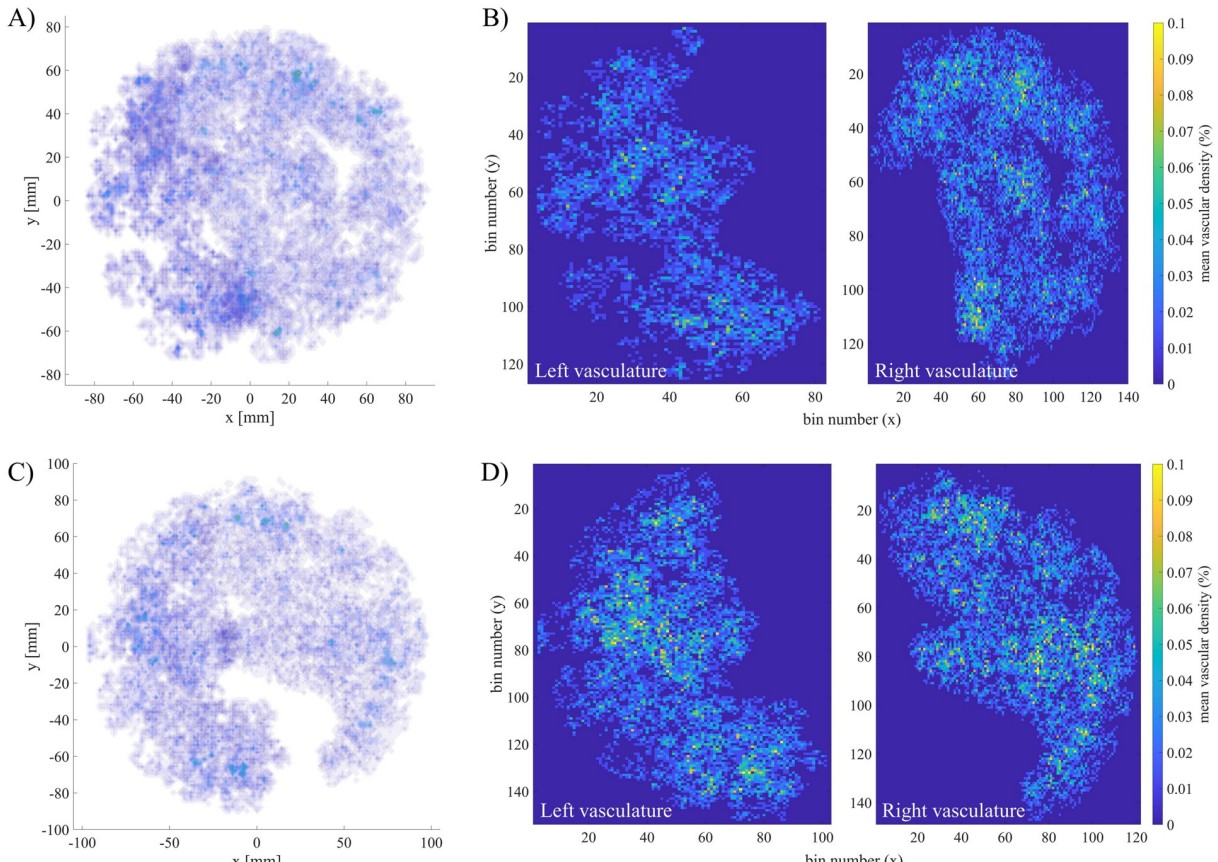

**Fig 13. Vascular density computed for feto-placental vasculatures at an isotropic voxel size of 116.5 µm.** Isosurface maps of nodal density and mean vascular maps are displayed for non-central (A,B) and centralised (C,D) umbilical cord insertion, respectively.

central region of the placentone is vessel-free. This is also observed in the second row of Fig 15, where the central region does not have associated nodal density. Moreover, vessels appear to spread less across the whole placentone. These features are associated with a smaller mean vascular density in comparison with the healthy tree (0.74±0.36 vs 0.84±0.32%). As expected, the FGR placentone yields lower mean path lengths (18.55±1.27 mm), lower terminal vessel diameters (0.018±0.006 mm vs 0.026±0.005 mm) and lower mean vascular density (0.20±0.09 vs >0.70±0.30%) when compared to the first two cases. This is also evident in Fig 15, where the FGR tree exhibits significantly reduced dimensions, associated with growth within a concentrated area of the placentone and a poorly perfused placentone periphery.

## Discussion

### A new generative model for controlled synthetic feto-placental vasculature creation

In this paper, we present a novel generative algorithm of feto-placental vasculature morphology. We create a pipeline characterised by: (1) its flexibility, allowing for the automated and user-controlled generation of tailored geometrical models of the full feto-placental vasculature or of its individual components; (2) the application of directly imposed morphological parameters (either obtained via medical imaging or determined by the user) which guide feto-

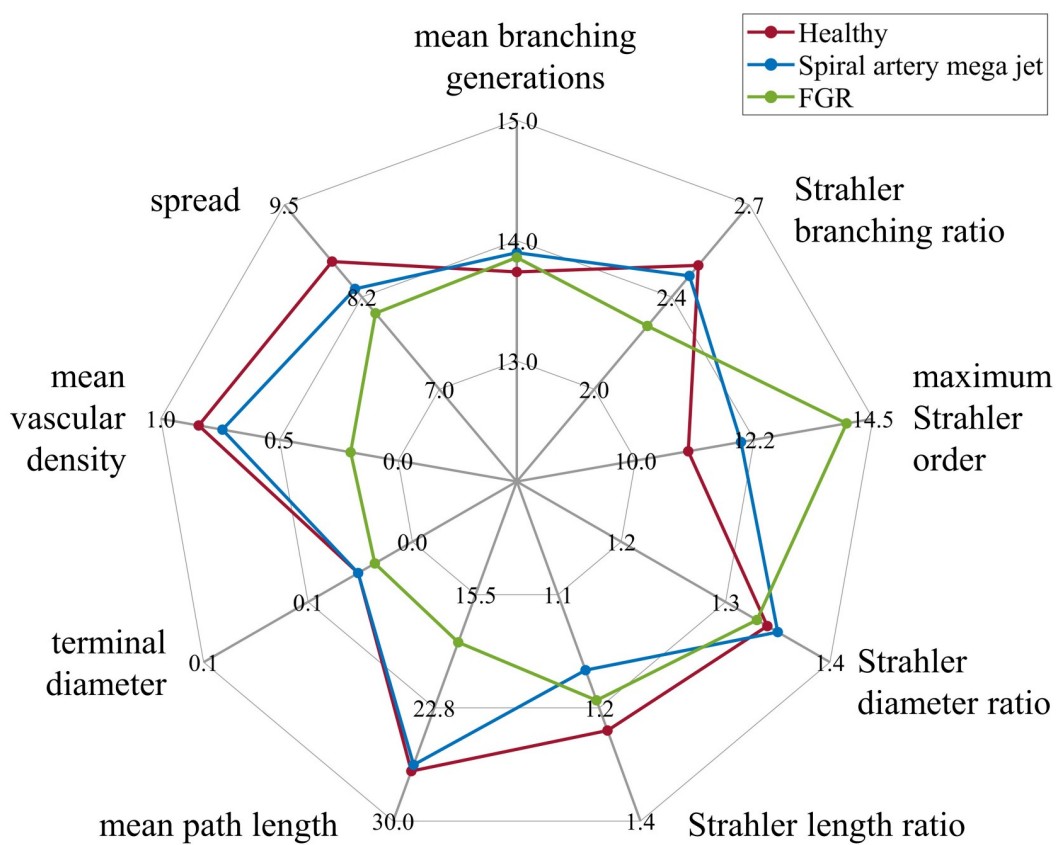

**Fig 14. Comparison of key mean topological metrics for three distinct placentones.** Metrics obtained for a healthy fetal tree, a fetal tree obtained with spiral artery mega jet and a fetal tree associated with fetal growth restriction are represented.

placental vasculature growth; (3) the dependence of generated vascular structural metrics on key parameters such as branching angles and length-to-diameter ratios, as determined by the global sensitivity analysis performed; (4) the stochastic nature of outputted structures, as evidenced by varying topological metrics between runs with the same input parameters; (5) the heterogeneity in spatial vascular density for generated feto-placental vasculatures.

Whilst we suggest appropriate parameter distributions for healthy and dysfunctional placentas at 35 gestational weeks, we do not claim that these are precise. However, they are reasonable estimates according to the current literature. We anticipate that parameters will be updated as new morphometric evidence emerges, or adjusted according to gestation or pathology. With the suggested parameter ranges, morphological and topological properties of the placental shape and feto-placental vascular structures generated with our model are comparable to *in vivo* and *ex vivo* measured properties, as well as *in silico* computational geometries (see Assessment of general topological properties and comparison with literature). We emphasize that these, particularly the *in silico* estimates, should not be considered absolute gold standards for validation but are provided simply for comparison. These structures are inherently stochastic, as generative parameters are controlled by Gaussian and uniform distributions resulting in parameters which vary at each branching generation and variability in topological metrics between algorithm runs.

As per the sensitivity analysis performed, the chorionic and villous vascular structures obtained and associated spatial distribution critically depend on the global distribution penalty

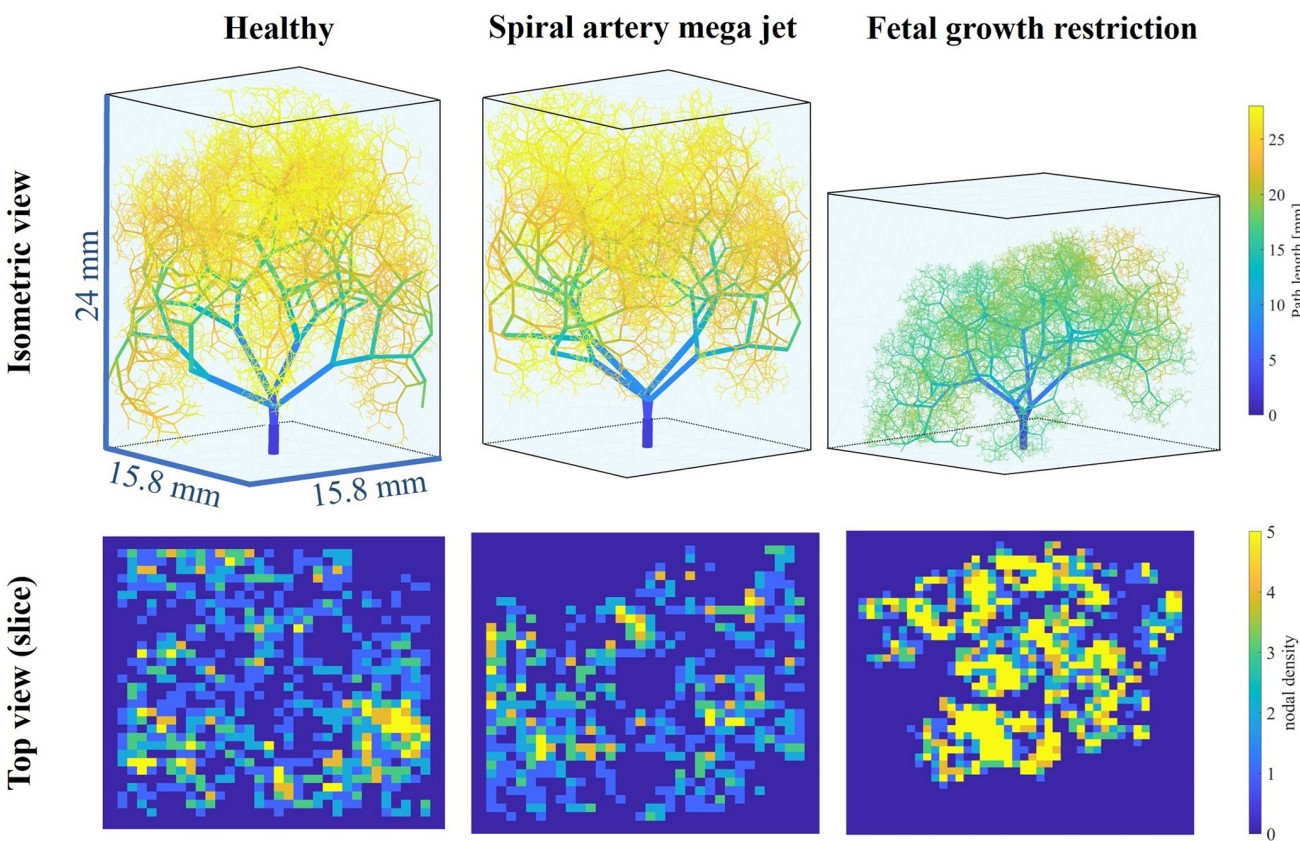

**Fig 15. Placentone fetal trees of up to 15 branching generations, created for three distinct topological cases.** Top row: path length (mm) for a healthy fetal tree, a tree in the presence of a spiral artery mega jet, and a tree in the presence of fetal growth restriction. Bottom row: axial slices of nodal density for all three fetal trees.

weights ($cf_1$, $cf_2$). This highlights how important these penalty weights are in promoting appropriate spatial distribution of vascular structures: a decrease in $cf_1$ (and therefore higher $cf_2$) yields vasculatures with less spatial dispersion and more oriented towards the basal plate, while higher $cf_1$ values allow for a greater maximisation of the distance between branches, associated with greater spread. The Supporting information also includes examples of placentone fetal trees (S2 Fig) and key topological metrics obtained for chorionic (S2 Table) and villous vessels (S5 Table) generated with different penalty weights. Branching properties and spread of chorionic and villous vessels are also highly dependent on the optimisation of branching angles. On the other hand, the mean stem diameter influences all topological metrics of generated villous trees. Both chorionic vessels and villous tree structures appear strongly asymmetric in branching, as given by Strahler branching ratios beyond 2.3. These computational results are similar to those obtained by a previous *in silico* study [20].

The structure of complete feto-placental vasculatures is affected by input parameter distributions and the location of umbilical cord insertion; however, branching statistics remain mainly unaffected, similarly to previous computational geometries imposing heterogeneous regional vascular densities [21]. Mean vascular densities varied between structures with non-central and centralised umbilical cord insertion, still within the ranges reported by Aughwane *et al.* for voxel size $\geq 116.5$ $\mu$m ($0.5 \pm 0.5\%$, range 0.3–1%). Higher vascular density was also observed in the inner regions of the placenta. This agrees with the results obtained by

Aughwane *et al.*, which show a downward trend of vessel density towards the periphery [27]. According to Byrne *et al.*'s *ex vivo* analysis of feto-placental vascular architecture, a non-central umbilical insertion is associated with a more unequal arterial supply (0.34:0.66 ratio) when compared to a centralised insertion (0.55:0.45 ratio) [21]. Our results showcase a similar trend (non-central insertion: 0.37:0.63 ratio; centralised insertion: 0.48:0.52 ratio). Moreover, Byrne *et al.* noted that vascular structures with a non-central insertion tend to exhibit greater variability in regional vascularisation compared to a centralised insertion, often presenting larger areas of lower density vasculature [21]. This is reflected by a higher coefficient of variation in vascular density (86.1 vs 76.9%). While our two feto-placental vasculatures showcase variability in regional vascularisation, this remains consistent regardless of whether the umbilical insertion is non-central or centralised, as showcased by similar coefficients of variation in vascular density (81.3 vs 82.1%). Thus, to further investigate the structural disparities among vascular configurations with different umbilical insertions, it would be necessary to generate a larger dataset of feto-placental vascular structures and assess their regional vascular density.

Our algorithm also supports dichotomous and monopodial branching in the generation of chorionic vessels, although there is no preference for either branching mode in feto-placental vasculatures with different umbilical artery insertions. It has been previously shown that a non-central umbilical insertion is associated with a higher number of monopodial chorionic branches, while a centralised umbilical insertion yields a chorionic architecture which is mostly dichotomous [51]. The implementation of preferential branching patterns associated with specific umbilical cord insertions remains an aspect to be improved in future code release versions.

We do not directly model vascular architectures that mimic specific clinical phenotypes; instead, topological changes on vascular structures are induced by modified morphological inputted parameters, such as in the case of the placentone fetal tree associated with FGR (see Section Placentone fetal trees). Previous research using micro-CT and corrosion casting techniques to quantify feto-placental vasculature established a connection between FGR and poorly vascularised villi associated with impaired vascular branching and density, typically leading to hypovascular regions in the placenta [21, 72]. Our results for modelled FGR showcase these altered vascular density patterns, with a higher number of branches in the inward placentone volume and a poorly perfused peripheral region.

## Comparison with other state-of-the-art methodologies

Recent frameworks for the generation of feto-placental vasculatures at a multi-scale level focus on area- or volume-filling approaches, which intend to "fill" a certain area or volume based on a heuristic generation of the feto-placental vasculature [20, 21, 32]. While these can generate feto-placental vasculatures ready for anatomically based modelling of the placenta, they lack the flexibility to directly control key morphological parameters. Our generative algorithm, on the other hand, is flexible, enabling the direct control of a range of morphological parameters and allowing us to generate a section of the feto-placental vasculature or the complete structure.

Like previous computational studies [20, 21, 32], feto-placental vessels are simplified in shape and assumed to be cylindrical tubes. Despite yielding topological and morphological metrics comparable to those reported in the literature, this feature will neglect the development of asymmetric blood flow profiles usually present in tortuous vasculatures [73]. Three-dimensional realistic geometries have been used in the simulation of blood flow [74] and/or solute transport [23] in the placenta; however, due to computational restrictions, these are typically limited to components of the vasculature, not accounting for the entire structure. We

also adopted a cuboid placentone definition as opposed to ellipsoidal or cylinder, following on the definition employed by Saghian *et al.* in a previous computational study [3]. This choice simplifies mesh surface generation and the mathematical definition of a central cavity in comparison with a more complex placentone shape. Moreover, it helps reduce computational costs associated with fetal tree generation and iterative branch-to-mesh intersection checks.

The feto-placental vascular structures generated follow statistical distributions guided by data from the literature (when possible), with algorithm stochasticity emerging from these. This means that the same set of inputted parameters can yield different network structures, guaranteeing a variable distribution of vessels within the placental volume compared to that obtained when using deterministic algorithms which impose this heterogeneity as a direct constraint [21]. This provides capability not available with previous computational algorithms for the feto-placental vasculature, underpinning future computational experiments on how initial settings affect the generated structures. Whilst stochastic, our algorithm can reproduce the same structure given the same set of user-defined parameters by fixing the random seed in MATLAB with its *rng* function, not comprimising reproducibility of vascular structures. Similarly to other computational approaches [20, 21, 32], we do not directly model the fundamental mechanical and chemical processes which drive feto-placental vascularisation throughout pregnancy. However, many of our algorithm choices mimic these processes indirectly. A computational model directly incorporating these would require additional data for validation, including transient data obtained from the quantification of feto-placental vasculatures throughout pregnancy, as well as a range of functional properties such as oxygen gradients in the IVS. Given the complexity of the feto-placental vascularisation throughout pregnancy, this remains an open challenge for the development of future computational models applied to the placenta.

In addition, whilst we do not directly model the venous vasculature, we account for the space to be occupied by it via branching generation constraints (see Section Computational implementation: Branch intersection checks). This represents an enhancement compared to prior volume-filling methods, where venous vessels are assumed to occupy the same three-dimensional space as arterial vessels, with double the radius. While such an approach produces vascular structures with appropriate topological and branching characteristics, it yields inaccurate vascular densities if not guided by medical imaging.

## Potential applications and future perspectives

Our computational pipeline is versatile. When combined with a blood flow model implementation (e.g. [20, 21, 32, 75, 76]), it enables the assessment of functional metrics associated with different feto-placental vascular structures. Our objective is to provide full flexibility in selecting input parameters, which will be better informed as imaging methods and as imaging data analyses improve, and by allowing the incorporation of morphometric data across spatial scales (e.g. *ex vivo* synchrotron computed tomography [77]). In fact, future quantitative assessments of the placenta will provide carefully drawn feto-placental vascular structures and associated sets of key morphological parameters. This will provide a biological counterpart for additional structural validation of the vasculatures generated with our algorithm. Incorporating additional morphometric data on vessel geometry (e.g. vessel cross-section) and potentially subdividing vessel segments to yield more complex structural shapes will enable pipeline refinement in upcoming iterations. Moreover, all feto-placental vascular structures presented and analysed in this paper are based upon an idealised placental shape. Upcoming iterations to the pipeline include the direct use of patient-specific placental surfaces obtained from medical imaging.

By using a range of input morphological parameters, different gestational ages and pathological conditions can be generated, offering valuable insights into both morphological and functional changes [6, 9, 10, 68, 78]. On the other hand, there is a range of clinical phenotypes of interest which are associated with alterations to the feto-placental vasculature. For example, poor maternal flow into the IVS yields reduced oxygen delivery to the villi, which leads to hyper-branching patterns as an adaptive response to increase fetal uptake [79, 80]. Whilst we do not model these patterns directly, we acknowledge that this is a desirable feature in future code releases (e.g. including additional user options to control branching patterns by decreasing/increasing the number of branching nodes in a vascular network).

In addition, our generative algorithm complements *in silico* techniques for the simulation of MRI signals (e.g. [81, 82]), linking observed changes in MRI signals to specific placental perturbations. For example, FGR, characterised by smaller placental shapes and geometric alterations in the feto-placental vasculature, has been associated with reduced placental MRI $T_2^*$ [83, 84] and diffusivity [84, 85].

## Conclusion

In this study, we introduce a novel generative algorithm for creating *in silico* placentas. This flexible algorithm allows for user-controlled parameters, enabling automated generation of tailored geometrical models of feto-placental vasculatures, both as individual components (placental shape, chorionic vessels, unique fetal tree in a placentone) and as a complete structure. We demonstrate the versatility of this framework through clinically-relevant examples and assess the morphological and topological properties of the generated vasculatures against *in vivo* and *ex vivo* measurements and *in silico* predictions. We highlight the importance of vessel length variations and branching angles in shaping placental vasculatures generated with our algorithm. Further, the pipeline generates spatially varying vascular structures typical of the placenta, crucial for reflecting the complexity of real placental vasculature. It could be used in future studies to study of the impact of various morphological parameters on feto-placental vasculature function, not only enabling investigations into regional variability in perfusion and vascular resistance but also facilitating the representation of pathological conditions with significant clinical implications.

While our current focus is on generating arterial networks, we acknowledge the challenges associated with modelling downstream vessels. We propose future developments, including an appropriate representation of venous placental circulation and capillary pathways. The flexibility of the algorithm and control over key morphological parameters offer exciting possibilities for more in-depth investigations into the structural and functional characteristics of placental vascular networks and their role in pregnancy-related complications.

## Methods

### Computational implementation: Algorithms

Algorithms 1 and 2 describe the generation of chorionic and villous vessels in detail.

```
Algorithm 1 Generation of chorionic vessels
while i ≤ bg_c & number of segments ≤ bn_c do
                ▷ Loop through undetermined segments
  Obtain list of current terminal nodes
  for terminal nodes c_t ∈ list do
    Assess branch termination
    if branch is not terminated then obtain parent node C_{i-1}
                ▷ Generation of daughters B_1, B_2
      while B_1 = [ ] and B_2 = [ ] do
```

```
          Use Eq 7 and defined k_c to obtain desired daughter branch diame-
ters d_1^i, d_2^i Obtain desired parent-daughter branching angles θ_{pd1}^i, θ_{pd2}^i
          Use Eq 27 to obtain desired ltd_c^i and derive daughter branch
lengths l_1^i, l_2^i
        for all candidate nodes n ∈ chorionic plate mesh do
          Use Eq 12 to compute l_f and θ_f
                ▷ In the case of daughter B_1
          Compute branching angle and Euclidean distance between cur-
rent node ∈ c_t and candidate node ∈ n
          if branching angle ∈ θ_f & Euclidean distance ∈ l_f then
            Add candidate node to saved nodes δ and store loss function
value in L (Eq 21)
          end if
                ▷ In the case of daughter B_2
          Compute branching angles between (1) current node ∈ c_t and
candidate node ∈ n and between (2) B_1 and candidate node
          Compute Euclidean distance between current node ∈ c_t and can-
didate node ∈ n
          if branching angle (1) ∈ θ_f & Euclidean distance ∈ l_f &
branching angle (2) ∈ [θ_{dd}(min), θ_{dd}(max)] then
            Add candidate node to saved nodes δ and store loss function
value in L (Eq 21)
          end if
          for all saved nodes ∈ δ do
            if candidate branch intersects other tree segments then test
next node
            else if list of saved nodes = [ ] then break
            else B_{1,2} = node ∈ δ
            end if
          end for
        end for
      end while
    end if
    Add nodes to Tree
  end for
  i = i+1
end for
```

**Algorithm 2** Generation of villous trees

```
while i ≤ bg_v do
                ▷ Loop through undetermined segments
  Obtain list of current terminal nodes
  for terminal nodes c_t ∈ list do
    Assess branch termination
    if branch is not terminated then obtain parent node C_{g-1} and parent-
parent node C_{g-2}
      Calculate n_v using Eq 14
                ▷ Generation of daughters B_1, B_2
      while B_1 = [ ] and B_2 = [ ] do
        Use Eq 7 and defined k_v to obtain desired daughter branch diame-
ters d_1^i, d_2^i
        Obtain desired parent-daughter branching angles θ_{pd1}^i, θ_{pd2}^i
        Obtain desired ltd_v^i and derive daughter branch lengths l_1^i, l_2^i
        Define spherical surfaces S_1^i, S_2^i of radius l_1^i, l_2^i) of uniformly
sampled nodes and split nodes using n_v
                ▷ In the case of daughter B_1
        for all candidate nodes n ∈ S_1^i do
          use Eq 12 to compute θ_f
```

```
              Compute branching angle between current node ∈ 𝒄ₜ and candi-
date node ∈ n
              if branching angle ∈ θ_f then
                Add candidate node to saved nodes δ and store loss function
value in L (Eq 23)
              end if
          end for
                  ▷ In the case of daughter B₂
          for all candidate nodes n ∈ S₂ⁱ
              use Eq 12 to compute θ_f
              Compute branching angles between (1) current node ∈ 𝒄ₜ and
candidate node ∈ n and between (2) B₁ and candidate node
              if branching angle (1) ∈ θ_f & branching angle (2) > θ_dd(min)
then
                  Add candidate node to saved nodes δ and store loss function
value in L (Eq 23)
              end if
          end for
          for all saved nodes δ do
              if candidate branch intersects other segments or outer domain
then test next node
              else if list of saved nodes = [ ] then break
              else B₁,₂ = node∈ δ
              end if
          end for
        end while
      end if
      Add nodes to Tree
  end for
  i = i+1
end while
```

## Computational implementation: Fine-tuning spatial distribution of feto-placental vasculature

Our algorithm incorporates global branch distribution penalties encoding tree properties that are used to influence the spatial distribution of segments in both the chorionic plate and the IVS. For both cases, these metrics are utilised to calculate a score for each candidate daughter node in the list. Nodes on the list are then assigned a rank, indicating the preferred nodes for selection. These scores are inspired by the vessel space-filling forces reported in Yang *et al.*, which aim to provide a realistic spatial distribution of vascular structures [86].

**Chorionic vessels.** The global penalty for each candidate daughter node is determined by two distances: its distance to other vascular nodes ($D_1$), which is maximised, and its distance to the chorionic plate centroid ($D_2$), which is minimised.

Remembering Eq 17 and assuming a list of candidate daughter nodes defined by $\delta_i = (X_i, Y_i, Z_i)$, we define

$$D_1 = \sqrt{\sum_{i=1}^{3}(\delta_i - C_i)^2} \quad \text{and} \quad D_2 = \sqrt{\sum_{i=1}^{3}(\delta_i - G_i)^2} \tag{20}$$

where $G_i$ is the chorionic plate centroid. By normalising $D_1$ and $D_2^{-1}$, we can then define scores **L** for all nodes in $\boldsymbol{\delta}$:

$$L(\delta_i) = cf_1 \cdot D_{1i} + cf_2 \cdot D_{2i}, \tag{21}$$

where $cf_1$ and $cf_2$ are weights. **L** is then sorted in descending order to define the overall ranking of candidate daughter nodes.

**Villous vessels.** Two main distances control the generation of new villi in the IVS, specifically the distance between the candidate daughter node and: (1) the plane characterised by normal $n_V$ ([Eq 14]) ($D_1$), which provides a spatial division between daughter segments and is maximised; (2) the basal plate ($D_2$), which is minimised.

Assuming the same notation for a tree and candidate daughter nodes as in subsection Chorionic vessels, and defining each parent-parent node as $C_{g-2} = (C_{g-2,1}, C_{g-2,2}, C_{g-2,3})$ we state

$$D_1 = \frac{|\delta_i \cdot A + \delta_i \cdot B + \delta_i \cdot C + I|}{\sqrt{A^2 + B^2 + C^2}} \quad \text{and} \quad D_2 = min\left(\sqrt{\sum_{i=1}^{3}(\delta_i - m_{bi})^2}\right), \tag{22}$$

where $I = -(A \cdot C_{g-2,1} + B \cdot C_{g-2,2} + C \cdot C_{g-2,3})$ and $m_{bi}$ are the basal plate mesh nodes. By normalising $D_1$ and $D_2^{-1}$, we can then define scores **L** for all nodes in $\boldsymbol{\delta}$:

$$L(\delta_i) = cf_1 \cdot D_{1i} + cf_2 \cdot D_{2i}, \tag{23}$$

where $cf_1$ and $cf_2$ are weights. **L** is then sorted in descending order to define the overall ranking of candidate daughter nodes.

## Computational implementation: Branch intersection checks

During the generation process of chorionic and villous vessels, intersection checks are conducted. Villous vessels are assessed for intersections with the outer domain mesh and other segments, while chorionic vessels are only evaluated for intersections with other segments. The evaluation and resolution process involves the following steps: (1) Checking intersections between segments and the placental mesh; (2) Defining a region of interest (*ROI*) for branch-to-branch intersection checks; (3) Performing a preliminary test for branch-to-branch intersections assuming infinite cylinders; (4) Conducting a complete test for branch-to-branch intersections assuming finite cylinders.

1. **Branch-to-mesh intersection**
   All candidate daughter nodes are tested on whether they are inside or outside a certain closed triangulated mesh (either the placentone geometry or the placental surface). This is done via a MATLAB pipeline which employs a mesh voxelisation algorithm to convert the triangulated mesh into a voxelised image. A ray intersection method is then used to determine the intersection points of rays and occupied cells in the image [87].

2. **Definition of region of interest**
   To detect branch-to-branch intersections within appropriate computational cost, only the branches in closest proximity to the candidate daughter branch are considered for branch-to-branch intersection tests. For each candidate daughter branch of length $l^i$, a spherical *ROI* of radius $(3/5)l^i$ is centered at the branch midpoint. This particular radius was determined through heuristic exploration of various scaling factors and visual inspection of the generated vascular structure. A scaling factor of $(3/5)$ (equivalent to a *ROI* diameter of $1.2l^i$) ensures that the *ROI* fully encapsulates the candidate daughter branch. This is mathematically sufficient to determine whether any branch-to-branch intersections occur, while keeping a suitable computational cost (a larger spherical *ROI* would encompass additional branches unnecessarily, resulting in increased computational times). Any vascular branches with start/end nodes inside the *ROI* are stored and tested for intersections with the candidate daughter branch (steps 3 and 4).

3. **Infinite cylinder testing**

   A preliminary branch-to-branch test assuming the segments as infinite cylinders is performed by computing the line-to-line minimum distance, $DL_{ij}$. For two segments $B_i$ and $B_j$ defined by start and end nodes $(S_i, E_i)$ and $(S_j, E_j)$, diameters $d_i$ and $d_j$, and direction vectors $\mathbf{u_i}$ and $\mathbf{u_j}$, $DL_{ij}$ is calculated as

   $$DL_{ij} = \begin{cases} \left| \dfrac{(\mathbf{u_i} \times \mathbf{u_j}) \cdot S_i S_j}{|\mathbf{u_i} \times \mathbf{u_j}|} \right|, & \text{if } |\mathbf{u_i} \times \mathbf{u_j}| \neq 0 \\[3mm] \left( \dfrac{|\mathbf{u_i} \times S_i S_j|}{|\mathbf{u_i}|} \right), & \text{otherwise.} \end{cases} \tag{24}$$

   The minimum $DL_{ij}$ required to prevent branch-to-branch intersection is equal to or above $\left( \frac{d_i}{2} + \frac{d_j}{2} \right)$, which includes branches in contact [81]. To account for empty space to be occupied by venous vessels, we adapt this threshold by including two venous branches with radii of $\frac{3}{2} \cdot \frac{d_i}{2}$ and $\frac{3}{2} \cdot \frac{d_j}{2}$, respectively [59], as stated in Model section Morphological metrics computed. The new threshold for all branches becomes:

   $$DL_{ij} \geq \left( \frac{d_i}{2} + \frac{d_j}{2} \right) + \frac{3}{2} \cdot \frac{d_i}{2} + \frac{3}{2} \cdot \frac{d_j}{2} = 5 \cdot (d_i + d_j) \tag{25}$$

   Since we want to avoid branch-to-branch contact and allow sufficient space for additional vessels, we assume

   $$DL_{ij} > 7 \cdot (d_i + d_j) \tag{26}$$

   If Eq 26 is respected, the candidate daughter branch is accepted; otherwise, the algorithm proceeds to step 4.

4. **Finite cylinder testing**

   The point coordinates and radius defining both segments are used to create 3D parametric surface cylinders. These are used as input to generate triangulated mesh surfaces using predefined MATLAB functions [54]. The surface meshes are then converted to convex meshes (i.e. meshes with all internal angles $< 180°$), since these are easier to represent and process in MATLAB for collision detection. These convex meshes are then tested for collision using a built-in function from the Robotics System Toolbox based on the Gilbert–Johnson–Keerthi (GJK) distance algorithm [88]. This algorithm computes the Euclidean distance between two convex shapes to detect if they intersect or collide. It starts with an initial simplex (the smallest shape containing the origin) which is iteratively updated by calculating the farthest points along the surfaces of the shapes in the direction of the vector between their respective centers. This process refines the approximation of the closest points between the shapes until convergence or until reaching a maximum number of iterations. At each iteration, the algorithm checks if the origin lies within the convex hull formed by the simplex: if it does, this indicates an intersection between the shapes, prompting algorithm termination.

   If no collisions are detected during checks 1–4, the candidate daughter branch is accepted.

## Computer specifications and algorithm performance

All computations and analysis related to chorionic vessels and placentone fetal trees were performed on a personal laptop with a processor Intel Core i7–10870H CPU @ 2.20GHz, 64 GB RAM memory and NVIDIA GeForce RTX 3080 Laptop GPU with 16 GB RAM. Computations related to complete feto-placental vasculatures were performed using super-computing facilities (Computer Science High Performance Computing Cluster, University College London), using 1 core with 64 GB RAM for each model run. The computational cost for generating vascular structures varied by type and input parameters. Creating placentone fetal trees (up to 15 branching generations, with 9,950 to 17,684 branches) took 2–4 minutes. For the chorionic vessels (up to 6 branching generations), the computational time was 8–12 minutes. The complete feto-placental vasculatures, with 929,157 and 849,085 branches for centralised and non-central insertion structures, respectively, took $\sim$ 2.5–3.5 days due to the increased structural complexity and numerous intersection checks. While execution times for previous fetoplacental vasculature generation algorithms are not currently available, our computational performance shows improvement compared to earlier published algorithms for cerebrovascular generation (e.g. $\sim$ 76 minutes for 4,565 branches) [89] and adaptive constrained constructive optimization methods for the synthetic vascularization of complex anatomies (e.g. $\sim$ 23 hours for 8,000 terminal branches) [90]. However, there is still room for improvement in future algorithm releases to reach computational times similar to the accelerated constrained constructive optimisation techniques aimed at enhancing algorithm performance (e.g. $\sim$ 39 seconds for 16,000 branches) [91].

## Tunable parameter definitions and suggested ranges

We consult the literature to estimate plausible statistical distributions for each parameter and sample parameters from these distributions to generate placental structures stochastically. We emphasise that our intention is that these parameter choices are flexible, and should be adjusted based on the desired morphometric statistics and user-dependent choices. This enables generation of healthy or dysfunctional placentas at different stages of gestation, depending on the input provided.

Tables 6–9 show all tuneable input parameters and assumptions for their definition, given our literature-based estimates, and provide plausible ranges for healthy and dysfunctional scenarios.

**Placental size/shape and cord insertion.**   Table 6 outlines our assumptions and plausible ranges for the parameters that control placental size, shape and cord insertion.

**Table 6. Tuneable algorithm input parameters for the generation of placental size and shape and cord insertion with suggested literature-based estimates.**

| | Placental size/shape and cord insertion | | |
|---|---|---|---|
| Parameter | Assumptions | Input: healthy | Input: dysfunction |
| $V$ | Uniform distribution [6] | $\sim$ Unif(400, 500) cm$^3$ within last ten weeks of gestation [39, 48, 64, 66] | $\downarrow$ 20–45% [6, 38, 39] |
| $r_{maj}$ | Normal distribution [37, 49] | $\sim \mathcal{N}(9.07, 0.181)$ cm post-delivery [37] | mean value $\downarrow$ 0.5–4% [92] |
| $E$ | Normal distribution [36] | $\sim \mathcal{N}(0.49, 0.17)$ [36] | mean value $\uparrow$ 0–8% [36] |
| $CCI$ | Normal distribution [36] | $\sim \mathcal{N}(0.36, 0.21)$ [36] | mean value $\downarrow$ 0–11 or $\uparrow$ 0–8% [36] |
| $mt$ | Relaxing $mt$ enables generation of placentas with marginal cord insertion (adverse fetal/maternal outcomes) ([68]) | 20 mm [68] | < 20 mm [68] |

**Table 7. Tuneable algorithm input parameters for chorionic vessel generation with suggested literature-based estimates.**

| Parameter | Assumptions | Input: healthy | Input: dysfunction |
|---|---|---|---|
| | | **Chorionic vessel generation** | |
| $ud$ | Normal distribution [62] | $\sim \mathcal{N}(4.6, 0.9)$ mm at 35 weeks of gestation [62] | mean value $\downarrow$ 20% [66] |
| $ltd_c$ | Normal distribution [20, 21] | $\sim \mathcal{N}(ltd_c(g), 1)$, see Eq 27 | $\uparrow$ 5–10% [32] |
| $k_c$ | Relaxing $k_c$ enables changes in the relationship between vessel diameter and branching generations | 3.2 at 35 weeks of gestation, inferred from data from [32] | increasing or decreasing $k_c$ yields larger or smaller vessel diameters |
| $\theta_{dd}$ | Uniform distribution [51] | $\sim$ Unif(70, 100)° [51] | $\downarrow$ 0–8% [93] |
| $\theta_{pd}$ | Uniform distribution inferred from $\theta_{dd}$ | $\sim$ Unif(35, 50)° | $\downarrow$ 0–8% |
| $a$ | Normal distribution [86] | $\sim \mathcal{N}(1, 0.05)$ [86] | sigma value $> 0.05$ ($\uparrow a$) |
| $bg_c$ | - | 6–8 [43, 51, 52] | - |
| $bn_c$ | - | 60–100 [43, 51, 52] | $\downarrow$ 45% [44] |
| $cf_1$ | - | 0.3–0.7 | - |
| $cf_2$ | - | 0.3–0.7 | - |

**Chorionic vessel generation.** Table 7 outlines our assumptions and plausible ranges for the parameters that control the generation of umbilical and chorionic vessels. The chorionic length-to-diameter ratio $ltd_c$ varies over branching generations. Control placenta data collected as part of the National Children's Study [32] has been used to derive a polynomial fit representing the length-to-diameter ratio ($ltd$) as a function of the branching generation ($g$):

$$ltd_c(g) = ag^3 + bg^2 + cg + d, \tag{27}$$

where $a$ is 0.02629, $b$ is -0.655, $c$ is 5.176 and $d$ is -0.7898.

**Placentone domain generation.** Table 8 outlines our assumptions and plausible ranges for the parameters that control placentone size and shape, as well as central cavity dimensions.

**Villous vessels generation.** Table 9 outlines our assumptions and plausible ranges for the parameters that control the generation of villous vessels. Post-delivery microscopic analysis of terminal villi [10] and image-based computational models of villous trees [21] show great variability in vessel branching angles. These range from 40 to 70° in terminal vessels [10] and are reported as a normal distribution in computationally generated villous trees [20, 21]. Normal distribution branching angle estimates from Clark *et al.* and Byrne *et al.* indicate elevated standard deviation (e.g. $\mathcal{N}(46.16, 30.20)°$). As such, we assume that parent-daughter branching angles can be characterised by a normal distribution, as showcased in Table 9.

Similarly to the chorionic vasculature, we expect the villous length-to-diameter ratio $ltd_v$ to vary over branching generations. Previous computational models of the placenta represent the overall placental $ltd$ as a normal distribution with elevated standard deviations (e.g.

**Table 8. Tuneable algorithm input parameters for placentone domain generation with suggested literature-based estimates.**

| Parameter | Assumptions | Input: healthy | Input: dysfunction |
|---|---|---|---|
| | | **Placentone domain generation** | |
| $V$ | Uniform distribution [6] | $\sim$ Unif(400, 500) cm$^3$ within last ten weeks of gestation [39, 48, 64, 66] | $\downarrow$ 20–45% [6, 38, 39] |
| $2t_{half}$ | - | 24 mm at 35 weeks of gestation [3, 6] | $\downarrow$ 25% [44] |
| $n_p$ | - | 60–100 at 35 weeks of gestation [3, 6, 43, 52] | $\downarrow$ 45% [44] |
| $CC_r$ | - | 1.88 at 35 weeks of gestation [3] | no change [3] |
| $CC_h$ | - | 5.6 at 35 weeks of gestation [3] | $\uparrow$ 100–300% [3] |

**Table 9. Tuneable algorithm input parameters for villous vessels generation with suggested literature-based estimates.**

| Villous vessels generation | | | |
|---|---|---|---|
| Parameter | Assumptions | Input: healthy | Input: dysfunction |
| $d_s$ | Normal distribution | $\sim \mathcal{N}(0.7, 0.03)$ mm at 35 weeks of gestation [43] | ↓ 20% [66] |
| $ltd_v$ | Normal distribution [20, 21] | $\sim \mathcal{N}(10.5, 1)$ | - |
| $k_v$ | Relaxing $k_v$ enables changes in the relationship between vessel diameter and branching generations | 3 at 35 weeks of gestation [43] | increasing or decreasing $k_v$ yields larger or smaller vessel diameters |
| $a$ | Normal distribution [86] | $\sim \mathcal{N}(1, 0.05)$ [86] | sigma value $> 0.05$ (↑ $a$) |
| $\theta_{pd}$ | Uniform distribution [20, 21] | $\sim \mathcal{N}(45, 20)°$ [10, 20, 21] | mean value ↑ 10–45% [10] |
| $\theta_{dd}$ | - | 25° | no change |
| $cf_1$ | - | 0.5–0.7 | - |
| $cf_2$ | - | 0.3–0.5 | - |
| $bg_v$ | - | 13–15 [20, 21, 43] | no change [44] |

$\mathcal{N}(15.87, 8.21)$ mm [20, 21]; This may be explained by the fact that these estimates represent the entire feto-placental vasculature, and therefore such a large standard deviation may not be appropriate for the generation of villous trees.

## Supporting information

**S1 Fig. Boxplots showcasing variability of additional key topological metrics obtained after 50 algorithm runs for the generation of a) chorionic vessels and b) villous vessels.**
(TIF)

**S2 Fig. Placentone fetal trees generated with different global branch distribution penalty weights while keeping other input parameters fixed.** a) Fetal tree generated with $cf_1 = 0.4$ and $cf_2 = 0.6$. b) Fetal tree generated with $cf_1 = 0.8$ and $cf_2 = 0.2$.
(TIF)

**S1 Table. Key topological metrics obtained for chorionic vessels generated with different seed numbers.** The original chorionic plate surface mesh (1251 seeds) yields chorionic vascular structures with hindered topological metrics (e.g. lower number of chorionic vessels and less branching generations), while the mesh with sub-triangulated elements (4951 seeds) gives rise to better vascular structures (e.g. higher number of chorionic vessels with increased spread).
(PDF)

**S2 Table. Key topological metrics obtained for chorionic vessels generated with different global distribution penalty weights ($cf_1$, $cf_2$).** Higher $cf_2$ values result in fewer chorionic vessels, along with reduced spread and smaller mean path lengths. This is the result of vessels being generated towards the centroid of the chorionic plate. Higher $cf_1$ values, on the other hand, promote greater vessel distribution across the chorionic plate, with an increased number of vessels and longer mean path lengths, though this can cause vessels to bypass the inner region of the chorionic plate. To achieve a balanced distribution of vessels while maintaining appropriate topological characteristics, a middle range of values is recommended (e.g. $cf_1 = 0.3–0.7$; $cf_2 = 0.3–0.7$).
(PDF)

**S3 Table. Key topological metrics obtained for chorionic vessels generated up to 8 branching generations with different tolerances in branch lengths ($tol_l$).** While smaller $tol_l$ values

are associated with lower computational times, they hinder vessel generation as highlighted by a smaller number of vessels and mean branching generations. In contrast, increasing $tol_l$ values lead to appropriate topological metrics, but at a much higher computational cost. Therefore, a middle range $tol_l$ (e.g. 1.5 mm, equivalent to 8–13% of chorionic vessel lengths) offers a balanced compromise between optimal topological metrics and computational efficiency.
(PDF)

**S4 Table. Key topological metrics obtained for chorionic vessels generated up to 8 branching generations with different tolerances in parent-daughter branching angles ($tol_\theta$).** While smaller $tol_\theta$ values are associated with lower computational times, they hinder vessel generation as highlighted by a smaller number of vessels. In contrast, increasing $tol_l$ values lead to a greater number of generated vessels, but at a much higher computational cost. Therefore, a middle range $tol_\theta$ (e.g. 0.2182 radians, equivalent to 25–35% of chorionic vessel parent-daughter branching angles) offers a balanced compromise between optimal topological metrics and computational efficiency.
(PDF)

**S5 Table. Key topological metrics obtained for villous vessels generated with different global distribution penalty weights ($cf_1$, $cf_2$).** $cf_1$ and $cf_2$ greatly affect the spatial distribution of vascular branches, as quantified by distances between vascular nodes and the basal plate (Distance 1) or between different vascular nodes (Distance 2). To obtain fetal trees with appropriate spatial dispersion, a middle range of values is recommended (e.g. $cf_1$ = 0.5–0.7; $cf_2$ = 0.3–0.5).
(PDF)

**S6 Table. Key topological metrics obtained for villous vessels with different tolerances in parent-daughter branching angles ($tol_\theta$).** While smaller $tol_\theta$ values are associated with lower computational times, they hinder vessel generation as highlighted by fewer branching generations and lower vascular density. This is also quantified by distances between vascular nodes and the basal plate (Distance 1) or between different vascular nodes (Distance 2). Increasing $tol_l$ values results in more branching generations, higher vascular density, and greater vessel spatial distribution, though at a higher computational cost. Therefore, a middle range $tol_\theta$ (e.g. 0.1178 radians, equivalent to 15% of villous vessel parent-daughter branching angles) offers a balanced compromise between optimal topological metrics and computational efficiency.
(PDF)

**S7 Table. List of abbreviations and parameters.**
(PDF)

**S8 Table. List of key parameter symbols used throughout the manuscript.** The general form is defined here, with some parameters being applicable to different scenarios (e.g. $k$ can be associated with the generation of chorionic or villous vessels, becoming $k_c$ or $k_v$, respectively).
(PDF)

## Author Contributions

**Conceptualization:** Diana C. de Oliveira, Kelly Payette, Jana Hutter, Lisa Story, Joseph V. Hajnal, Daniel C. Alexander, Rebecca J. Shipley.

**Data curation:** Diana C. de Oliveira.

**Formal analysis:** Diana C. de Oliveira.

**Funding acquisition:** Daniel C. Alexander, Rebecca J. Shipley.

**Methodology:** Diana C. de Oliveira, Hani Cheikh Sleiman, Kelly Payette, Paddy J. Slator.

**Project administration:** Jana Hutter, Lisa Story, Joseph V. Hajnal, Daniel C. Alexander, Rebecca J. Shipley.

**Software:** Diana C. de Oliveira, Hani Cheikh Sleiman, Paddy J. Slator.

**Supervision:** Jana Hutter, Joseph V. Hajnal, Daniel C. Alexander, Rebecca J. Shipley, Paddy J. Slator.

**Visualization:** Diana C. de Oliveira, Paddy J. Slator.

**Writing – original draft:** Diana C. de Oliveira, Paddy J. Slator.

**Writing – review & editing:** Diana C. de Oliveira, Hani Cheikh Sleiman, Kelly Payette, Jana Hutter, Lisa Story, Joseph V. Hajnal, Daniel C. Alexander, Rebecca J. Shipley, Paddy J. Slator.

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
