## [Decision Letter · Decision Letter 0]

25 Mar 2024

Dear Dr C. de Oliveira,

Thank you very much for submitting your manuscript "A flexible generative algorithm for growing <in silico=""> placentas" for consideration at PLOS Computational Biology. As with all papers reviewed by the journal, your manuscript was reviewed by members of the editorial board and by several independent reviewers. The reviewers appreciated the attention to an important topic. Based on the reviews, we are likely to accept this manuscript for publication, providing that you modify the manuscript according to the review recommendations.</in>

Sincerely,

Alison Marsden

Academic Editor

PLOS Computational Biology

Pedro Mendes

Section Editor

PLOS Computational Biology

Reviewer's Responses to Questions

**Comments to the Authors:**

Reviewer #1: This paper introduces a novel generative algorithm for creating in silico placentas. One of the advantages of the proposed algorithm is that it allows direct control over morphological parameters, a limitation of previous volume-filling algorithms.

There are some details relating to the validation of the algorithm that could be improved.

METHODS

M.1. What is the rationale for the tolerance choice in the generation of the chorionic and villous vessels?

M.2 The description of some of the rules for tree generation lacks a reference or rationale. For example,

Line 738: To account for empty space to be occupied by venous vessels, we assume that if DLij > 7/2 (di + dj ), the generated daughter branch is accepted; otherwise, the algorithm proceeds to step 4.

Or

Line 729: The ROI is defined as a sphere of radius (3/5)ld,

Can the authors explain what this is doing exactly:

Line 744: The surface meshes are then converted to convex meshes, which represent collision geometries, and tested for collision using built-in functions from the Robotics System Toolbox.

M.3. The point candidates for branching seem to be bounded to the initial mesh defined for the placental surface. How is this mesh generated? Does the point density (i.e., number of point candidates) affect the model's output? If so, how does it compare to the variabilities shown in Figure 11?

M. 4. Why is the placentone defined as a cuboid? An ellipsoidal or circular cylinder seems a more appropriate choice to describe this structure.

RESULTS

R.1. Some parameter ranges from the literature are missing in Table 4.

R.2 About half of the parameters presented in Table 4 to evaluate the morphology of the generated trees are from in-silico studies (mostly one study; the other in-silico study mentioned is a quantification of micro-CT images). In the study from Clark et al. 2015, the authors use for validation, the placental vasculatures are generated using a volume-filling algorithm. According to the authors, the limitations of the volume-filling algorithm are precisely what motivates the generative algorithm presented in this paper. Hence, it seems contradictory to use results from the implementation of Clark et al. 2015 as validation for their tree generation algorithm. It is this reviewer's opinion that the authors should limit validation to in vivo or ex vivo data.

R.4. What is the inner polygon of Figure 12 (and 14) representing?

R.5. The results highlight the crucial role of cf1 for the branching angle outputs. However, this metric is not reported in the literature, and it is provided by the authors to the model. How are the ranges reported in Table 1 defined? This is also not included in the discussion.

R.6. Can the authors provide additional details and interpretation of the results obtained regarding the "variability in output metrics from multiple algorithm runs"? What is exactly plotted in figures a) and b)? Is this divergence acceptable?

R.7. Not all the output variables included in Figures 7, 8, 9, and 10 seem to be included in Table 4. Also, the name used is different in some cases, which adds confusion.

Other comments:

Figure 7and 9, add label in the color scale

Image resolution of Figure 7, 8, 9,10, 11 and 13 is not optimal.

Line 322: missing word (Section")

Lines 349: missing words (more details in Sections "and")

Line 631: missing word (Section")

Reviewer #2: The manuscript presents a flexible generative algorithm for creating synthetic feto-placental vasculature (arterial trees so far) with biophysical models and user-defined control. Compared with previous generating algorithms, a substantial advance of the algorithm presented herein is to allow direct control of the vasculature morphological parameters (e.g. vessel dimensions, branching angles) for designated feto-placental network characteristics reflecting healthy or pathological feto-placental structures, with a notable feature of stochastic variability for a given set of input parameters. The algorithm itself is introduced in detail with clear physiological underpinning and demonstrated through clinically-relevant examples, together with comprehensive quantification and sensitivity analysis. Overall, this is a well-motivated in silico framework and offers a new avenue for investigating the structure-function relationships in the placenta. With further developments to incorporate venous trees and complete capillary pathways, the framework can pave the way for more efficient organ-scale placenta modeling and inform diagnosis of placental disorders in the future. However, certain aspects of the manuscript need to be improved to enhance its clarity and impact, which are detailed below.

1. The stochastic nature of the algorithm is a desired feature to reflect biological variability but may also impose challenges on reproducibility. Does the algorithm have a mechanism (e.g. random seed generator) to reproduce a designated vasculature generated with a given set of user-defined parameters?

2. The authors should also discuss the implications of the modelled stochasticity in a biological context as the realistic vascularisation process of the feto-placental network is unlikely to be purely random but rather orchestrated through mechanical/chemical signals during pregnancy. Also, it would be nice if the authors could clarify how the level of stochasticity-induced structural variation compares to the absolute sample differences associated with pathologies versus physiology.

3. Does the algorithm support both dichotomous branching and monopodial branching (as shown in the Fig 2b schematic)? And if yes, does it have a preference for either branching mode in different scenarios, say non-central v.s. centralised umbilical artery insertion? I suspect that unlike the predicted outcome in Fig 12c (only marginal difference in the mean vascular density but not other topological metrics), the structural differences for non-central and centralised insertion in real placentas may be more severe? Or are the topological metrics adopted here robust enought to tell the structural difference?

4. The whole feto-placental vasculature or placentone villous tree is generated using morphological parameters known a priori in literature, but there seems no validation or evaluation of the generated in silico replica against the biological counterpart where the morphological parameters were drawn? Is there a way to achieve such validation at this stage without functional inspection, which I understand will need blood flow simulations in the generated network (therefore out of this paper’s scope) to compare with clinical imaging data (e.g. flow rate, oxygenation measurements)? It would also be nice to discuss how clinical phenotypes have been reproduced, e.g. hypo/hyper-branching patterns.

5. More details and analyses about the healthy/pathological placentone fetal trees in Fig 15 should be given to showcase the potential workflow from medical imaging data to in silico placenta.

Some figures and text also need to be polished:

Fig 1, the zoomed-in are (black box) in (a) does not quite correspond to the lobule detailed in (b). Also in (b), may use normal arrows to replace the dot-line pointers for indicating the flow directions.

Fig 3, the use of numbered sections in the diagram is an issue as the sections in the main text are not numbered.

Fig 5, consider modifing the diagram flow to better reflect the decision tree from step 3 to step 6.

Fig 6, the feto-placental vasculature is not fit to size and somehow mispositioned in the coordinate system.

Figs 8, 10-11, the line plots are blurred when printed out. Consider a darker line colour. Also in the captions of Figs 8 & 10, “Clusters... associated with higher mu* and lower sigma values, are circled in black,” is “lower sigma values” the case?

Figs 12c, 14, explain the base coordinates of the radar plots, which are sometimes non-zero.

Double check all figure captions and make sure the concerned symbols are explained.

Currently, the main text is quite dense and sometime difficult to navigate due to a certain level of redundancy. The text may benefit from streamlining the contents in the Model, Results and Method sections.

Table 3, double check “settings from 6-13.” It is unclear to me how the parameters in table 3 lead to a round (rather than oval) chorionic plate in Fig 12a, b.

Page 18 line 454, provide references for “consistent with previous reports.”

Some cross-references are not working, e.g. missing section titles on page 13 lines 316 & 322, page 14, line 350, page 30 line 631.

**Have the authors made all data and (if applicable) computational code underlying the findings in their manuscript fully available?**

Reviewer #1: Yes

Reviewer #2: Yes

PLOS authors have the option to publish the peer review history of their article (what does this mean?). If published, this will include your full peer review and any attached files.

Reviewer #1: No

Reviewer #2: **Yes: **Qi Zhou

Figure Files:

Data Requirements:

Reproducibility:

References:

---

## [Decision Letter · Decision Letter 1]

30 Jul 2024

Dear Dr C. de Oliveira,

Thank you very much for submitting your manuscript "A flexible generative algorithm for growing in silico placentas" for consideration at PLOS Computational Biology. As with all papers reviewed by the journal, your manuscript was reviewed by members of the editorial board and by several independent reviewers. The reviewers appreciated the attention to an important topic. Based on the reviews, we are likely to accept this manuscript for publication, providing that you modify the manuscript according to the review recommendations.

The manuscript is almost ready, but we need you to address this issue: "The authors should include quantitative data supporting the heuristic determination of parameters and parameter ranges in the supplementary". Of course you should try to address other issues in the attached reviews as much as possible.

Sincerely,

Pedro Mendes

Section Editor

PLOS Computational Biology

Reviewer's Responses to Questions

**Comments to the Authors:**

Reviewer #1: The authors have provided additional details on the methods, which facilitate understanding their approach.

The authors should include quantitative data supporting the heuristic determination of parameters and parameter ranges in the supplementary (e.g., tolerances, number of seeds, cf1, and cf2 …). These parameters have a critical role in successfully generating a physiological placental vasculature; supporting information on how they are determined will be essential for reproducibility.

On the topic of validation, this reviewer finds that validation with an in silico-generated vascular model may be insufficient. This is especially true in this case since the motivation for this new algorithm, according to the authors, is to address the limitations of previous in silico placenta vasculature generation algorithms.

References to the model's ability to represent psychological vascular trees should be limited to references that provide histological or image quantification of ex vivo data.  For example, the sentence:" Both chorionic vessels and villous tree structures are strongly asymmetric in branching, as given by Strahler branching ratios beyond 2.3, within the range of those presented in the literature [20]" is presented as a fact, while the citation refers to a modeling study. In such cases, the authors should only claim that their model produces results similar to those of the cited research, making it clear to the reader that both are computational results.

Although computational cost is mentioned several times in the manuscript, no quantitative data is provided in this respect. Please include a short comment referring to the hardware employed and the compute time required to generate the full vascular tree. If available, compare against previously published algorithms for vascular generation.

Reviewer #2: The authors have satisfactorily addressed my comments and the manuscript is significantly improved in this revision. I would recommend publication with an optional suggestion for the authors to consider: adopt consistent axial scales between the gridlines for each parameter in Figs. 12 and 14 to facilitate date interpretation.

**Have the authors made all data and (if applicable) computational code underlying the findings in their manuscript fully available?**

Reviewer #1: Yes

Reviewer #2: Yes

PLOS authors have the option to publish the peer review history of their article (what does this mean?). If published, this will include your full peer review and any attached files.

Reviewer #1: No

Reviewer #2: No

Figure Files:

Data Requirements:

Reproducibility:

References:

---

## [Editor Report · Decision Letter 2]

6 Sep 2024

Dear Dr C. de Oliveira,

We are pleased to inform you that your manuscript 'A flexible generative algorithm for growing in silico placentas' has been provisionally accepted for publication in PLOS Computational Biology.

Best regards,

Pedro Mendes

Section Editor

PLOS Computational Biology

---

## [Editor Report · Acceptance letter]

24 Sep 2024

PCOMPBIOL-D-24-00315R2 

A flexible generative algorithm for growing in silico placentas

Dear Dr C. de Oliveira,

I am pleased to inform you that your manuscript has been formally accepted for publication in PLOS Computational Biology. Your manuscript is now with our production department and you will be notified of the publication date in due course.

With kind regards,

Zsofia Freund
